# Intrinsic adaptive value and early fate of gene duplication revealed by a bottom-up approach

**Guillermo Rodrigo[1,2]\*, Mario A Fares[1,2,3]†**

[1]Instituto de Biología Molecular y Celular de Plantas, CSIC – UPV, Valencia, Spain; [2]Instituto de Biología Integrativa y de Sistemas, CSIC – UV, Paterna, Spain; [3]Trinity College Dublin, University of Dublin, Dublin, Ireland

**Abstract** The population genetic mechanisms governing the preservation of gene duplicates, especially in the critical very initial phase, have remained largely unknown. Here, we demonstrate that gene duplication confers per se a weak selective advantage in scenarios of fitness trade-offs. Through a precise quantitative description of a model system, we show that a second gene copy serves to reduce gene expression inaccuracies derived from pervasive molecular noise and suboptimal gene regulation. We then reveal that such an accuracy in the phenotype yields a selective advantage in the order of 0.1% on average, which would allow the positive selection of gene duplication in populations with moderate/large sizes. This advantage is greater at higher noise levels and intermediate concentrations of the environmental molecule, when fitness trade-offs become more evident. Moreover, we discuss how the genome rearrangement rates greatly condition the eventual fixation of duplicates. Overall, our theoretical results highlight an original adaptive value for cells carrying new-born duplicates, broadly analyze the selective conditions that determine their early fates in different organisms, and reconcile population genetics with evolution by gene duplication.

DOI: https://doi.org/10.7554/eLife.29739.001

**\*For correspondence:**
guillermo.rodrigo@csic.es

†Deceased

**Competing interests:** The authors declare that no competing interests exist.

## Introduction

Gene duplication has enthralled researchers for decades due to its link to the emergence of major evolutionary innovations in organisms of ranging complexity (*Ohno, 1970*). The key aspect to deeply understand this process concerns the early stage, when the fate of the new-born gene is decided (*Innan and Kondrashov, 2010*). A classical theory predicts the fixation of duplicated genes in the population under neutral selective conditions (i.e. by random genetic drift; *Kimura, 1983*; *Lynch and Conery, 2003*). Hence, the loss of the new-born gene is the most common evolutionary fate. Once a duplicate is fixed, it is generally accepted that genetic redundancy leads to relaxed selection constraints over one or both gene copies, which increases the load in mutations (*Lynch and Conery, 2000*; *Keane et al., 2014*). In rare occasions, this evolutionary process leads to the origin of a novel, previously unexplored function by one of the gene copies (*Conant and Wolfe, 2008*).

However, because gene duplication can impose a cost to the cell by requiring additional resources for expression (*Wagner, 2005*; *Lynch and Marinov, 2015*; *Price and Arkin, 2016*), especially in simple organisms, purifying selection could preclude that fixation. Gene duplication can also unbalance tightly regulated pathways that are instrumental for the cell (*Papp et al., 2003*; *Birchler et al., 2005*), leading to diseases in complex organisms (*Tang and Amon, 2013*). A possible rationale that has been long recognized is that those duplicated genes that were fixed in the population immediately contributed with an adaptive value to the organism (*Innan and Kondrashov, 2010*). Even

though, it is still stunningly unclear to what extent natural selection could also take part in the process that drives the fixation, and also initial maintenance, of duplicated genes according to population genetics (*Lynch, 2007*).

Two basic hypotheses have been proposed to explain the selective advantage of duplicated genes. First, a higher gene expression level resulting from duplication could be favorable (*Riehle et al., 2001*). This hypothesis requires that the ancestral system (pre-duplication) is far from the optimal operation point; as far as to assert that nearby 100% expression increase is beneficial. This seems plausible in extreme circumstances, but not in routine environments for which the organism should be adapted (*King and Masel, 2007*). It is then not surprising that many of the reported examples in which a greater gene copy number is favorable relate to sporadic, mainly stressing environments (*Riehle et al., 2001*; *Gonzalez et al., 2005*). Arguably, if a duplicate were fixed in one of these environments, it would be rapidly removed by purifying selection once the extreme circumstance ceased. Moreover, beneficial single-point mutations occurring in the *cis*-regulatory region of the gene of interest would be mostly sufficient to face several environmental changes (*Wray, 2007*). Thus, this model is insufficient to clarify the origin of most duplications, although it could explain some particular cases.

Second, the functional backup provided by the second gene copy upon duplication may allow the rapid accumulation of beneficial mutations, either to develop a novel function (*Zhang et al., 1998*; *Bergthorsson et al., 2007*), or to escape from the conflict of optimizing alternative functions (*Hittinger and Carroll, 2007*; *Des Marais and Rausher, 2008*). The positive selection of these mutations may of course occur, as suggested by the dN/dS values (>1) reported for different genomic sequences (*Han et al., 2009*; *Fischer et al., 2014*). This requires, nevertheless, that the frequency of cells carrying a second gene copy in the population increases to a point at which a mutation in the duplicate is likely to be found; a condition that is not met during the critical very initial phase following duplication (*Lynch et al., 2001*). Therefore, such adaptive processes, although important for the long-term maintenance of duplicates, do not contribute much to increase their fixation probabilities.

In addition to these two hypotheses, it has been proposed that gene duplication could allow compensating for errors in the phenotypic response due to a loss of expression caused by genotypic or phenotypic mutations (*Clark, 1994*; *Nowak et al., 1997*; *Wagner, 1999*). This model needs to invoke high error rates to have an impact at the population level from the beginning, and then to reach prevalence of genotypes with duplication by overcoming genetic drift. Errors in phenotype could also be caused by stochastic fluctuations in gene expression (*Elowitz et al., 2002*; *Balázsi et al., 2011*), with gene duplication eventually reducing the amplitude of such fluctuations (*Kafri et al., 2006*; *Lehner, 2010*; *Rodrigo and Poyatos, 2016*). But this strategy works on average, that is, duplication may warrant more accuracy when multiple decisions in gene expression are considered. Thus, it is not obvious whether an individual (or some) with duplication is able to invade a population, especially in a fluctuating environmental context. This is a key, largely unexplored question that may preclude the support of this idea. Other mechanistic models have been proposed beyond the demand for increased expression or the accumulation of beneficial mutations (*Innan and Kondrashov, 2010*), yet do not convincingly resolve the main population genetic dynamical issue.

In this work, we tested the idea of error buffering to reveal the adaptive value that gene duplication has per se. Subsequently, we developed a comprehensive model to explain the early fate of duplicates compatible with population genetics (*Lynch et al., 2001*; *Lynch, 2007*), global gene expression patterns (*Qian et al., 2010*; *Gout and Lynch, 2015*; *Cardoso-Moreira et al., 2016*; *Lan and Pritchard, 2016*), and unexpected gene copy number variation rates (*Reams et al., 2010*; *Schrider et al., 2013*). To this end, instead of performing a conventional sequence analysis (top-down approach), we followed a very precise quantitative framework, based on biochemistry, to study the goodness of having a second gene copy for the cell without functional divergence (bottom-up approach). Using a gene of *Escherichia coli* (*lacZ*) as a model system from which to apply our theory, we showed, without loss of generality, that the sum of two different, partially correlated responses allows reducing gene expression inaccuracies (*Rodrigo and Poyatos, 2016*); inaccuracies that are a consequence of the inherently stochastic nature of all molecular reactions underlying gene expression (*Raser et al., 2004*; *Carey et al., 2013*) and suboptimal gene regulation (*Dekel and Alon, 2005*; *Price et al., 2013*). Here, we considered intrinsic and extrinsic noise sources (*Elowitz et al., 2002*), that is, stochastic fluctuations that are specific of a gene and fluctuations that are unspecific, so gene duplication is expected to only buffer intrinsic fluctuations. In turn, cell fitness

can weakly increase on average, if such errors in gene expression are costly (*Wang and Zhang, 2011*); that is, a stochastic fluctuation may take the system far from the optimal operation point if the system is deterministically centered in this point), and then genotypes with duplication can be fixed in the population. We further studied the genetic and environmental conditions that are more favorable for the selection of gene duplication.

## Results

### Quantitative biochemical view of a fitness trade-off

In cellular systems, fitness trade-offs arise because beneficial actions involve costs. Fitness is a complex figure integrating multiple components, so the enhancement of one component (vital attribute) usually comports the reduction of another component (e.g. stress resistance vs. reproductive success; *Casanueva et al., 2012*). This is critically revealed when the environment changes, as the relevance of each component mostly depends on the external conditions. Such components can be described in different ways according to the problem. A paradigmatic and simple fitness trade-off emerges when a given enzyme needs to be expressed to metabolize a given nutrient present in the environment (*Figure 1a,b,c*). On the one hand, the cell growth rate (here taken as a metric of fitness; *Elena and Lenski, 2003*) increases as long as the enzyme metabolizes the nutrient. On the other hand, the enzyme expression produces a cost to the cell (i.e. reduces its growth rate). Therefore, the enzyme expression needs to be very precise to warrant an optimal or near-optimal behavior (cost-benefit analysis). To solve this issue, regulations (mainly transcriptional) evolved to link enzyme expression inside the cell with nutrient amount available in the environment. An example of this paradigmatic system is the well-known lactose utilization network of *E. coli* (*Jacob and Monod, 1961*), where lactose (nutrient, environmental molecule) activates, through inhibition of LacI (transcription factor), the production of LacZ (enzyme). We used this model system to apply a theoretical framework (see Materials and methods) in order to reveal the intrinsic adaptive value of gene duplication under a fitness trade-off, as this system has been quantitatively characterized (*Dekel and Alon, 2005*; *Kuhlman et al., 2007*; *Eames and Kortemme, 2012*).

Cell fitness increases monotonically with lactose dose (following a Michaelis-Menten kinetics), but presents an optimum with LacZ expression (*Figure 1d*). This is because lactose does not introduce a cost into the system, but LacZ does. Here, we simply considered a cost function based on LacZ expression (i.e. more expression, more cost), with a marginal cost of 0.036 in the units of the model (*Dekel and Alon, 2005*). However, it would be more precise to have a cost function based on lactose permease (LacY) activity (*Eames and Kortemme, 2012*), another gene in the *lac* operon in charge of the uptake, rather than on LacZ expression. The regulation of the system appears to be quite accurate, as the actual and optimal dose-response curves roughly match (*Figure 1e*). By generating different dose-response curves with values of $x_0$ (lactose $EC_{50}$ on LacZ) between 0.01 and 1 mM, we found that most of them deviate from the optimal one (p = 0.02; Euclidean distance as a metric). This entails great phenotypic plasticity of the cell to cope with lactose variations. However, plasticity is not equal for all environmental changes. Whilst the system (in terms of LacZ expression or cell fitness) reaches optimal sensitivity at intermediate doses, it is quite insensitive at very low or very high doses, where lactose-LacZ information transfer falls down (*Figure 1f*).

### Gene duplication helps to better resolve the fitness trade-off

The LacZ expression in *E. coli* involves a variety of noisy actions, such as the LacI expression, the LacI-DNA binding, the RNA polymerase-DNA binding, and the transcriptional elongation process (*Elowitz et al., 2002*; *Raser et al., 2004*; *Carey et al., 2013*). The resulting stochastic fluctuations in expression can have an impact on fitness (*Figure 2*). Using a simple mathematical model, we simulated the stochastic LacZ expression of the wild-type system for a varying lactose dose (*Figure 3a, b*). The magnitudes of the stochastic fluctuations were chosen as to end in typical variations of lactose $EC_{50}$ of 10–100%, up or down, resulting in values of gene expression noise, around 0.5, compatible with experimental results (*Elowitz et al., 2002*). At a given dose, these simulations would correspond to different single-cell responses. We also considered a system with two copies of the *lacZ* gene, with total expression equal to the previous one-copy system, and simulated its stochastic response (*Figure 3c*). For the moment, we ensured gene dosage sharing to evaluate in a

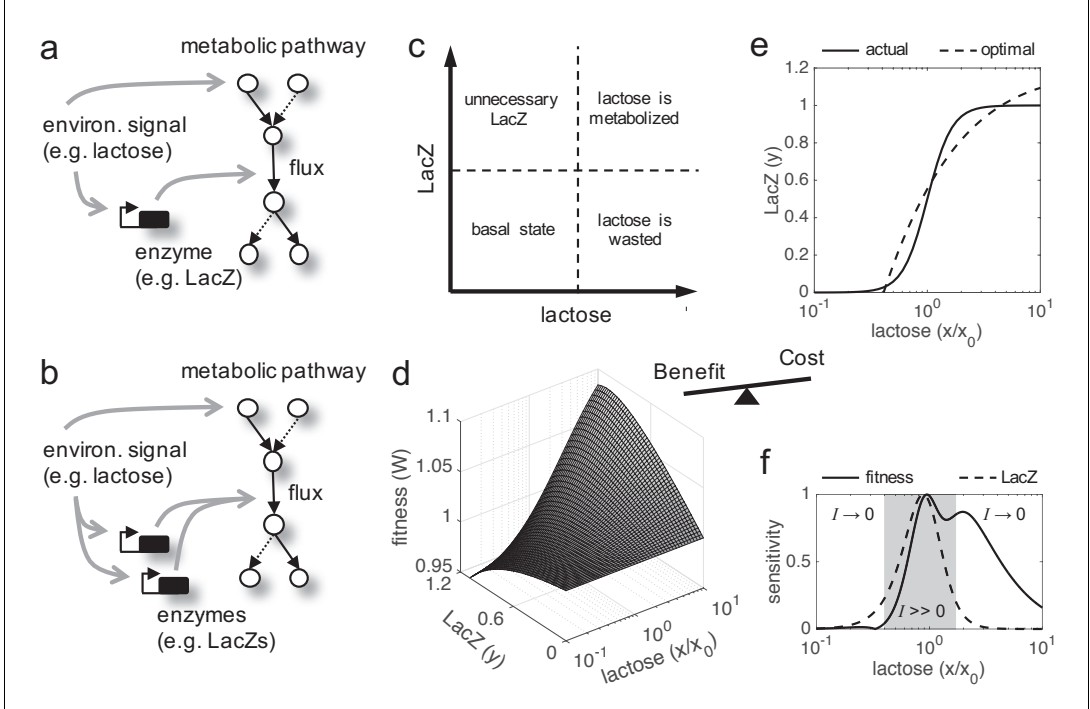

**Figure 1.** Fitness trade-off related to metabolic benefit and expression cost. (**a**) Scheme of a paradigmatic genetic system, coupling regulation and metabolism, where a given environmental signal determines the physiology of the cell. The environmental molecule can be metabolized by the cell, and it can also activate transcriptionally the expression of enzymes. A particular case is the lactose utilization system of *E. coli*. (**b**) Scheme of the same system with gene duplication. (**c**) Illustrative chart of the fitness trade-off showing four different cellular regimes. When the signal molecule (lactose) is not present in the medium, the expression of the enzyme (LacZ) is not required. However, when the signal molecule is present, the enzyme is required for its metabolic processing. (**d**) Fitness (*W*) landscape as a function of lactose (contributing to the benefit, *x* denotes its concentration) and LacZ (contributing to both the benefit and the cost, *y* denotes its concentration). This was experimentally determined. $x_0$ denotes the lactose $EC_{50}$ on LacZ expression, so $x/x_0$ is a normalized lactose concentration. (**e**) Dose-response curve between lactose and LacZ. The solid line corresponds to the actual regulation (experimentally determined), whilst the dashed line corresponds to a hypothetical optimal regulation (obtained by imposing $dW/dy = 0$). (**f**) Sensitivity to changes in lactose dose, either in fitness ($dW/dx$, solid line) or in LacZ ($dy/dx$, dashed line), characterizing the nonlinear phenotypic plasticity of the cell. Each curve is normalized by its maximum. This also measures sensitivity to molecular noise. The region where information transfer is high is shaded.

DOI: https://doi.org/10.7554/eLife.29739.002

quantitative way the goodness of having a second gene copy for the cell without invoking the need for more expression. We observed that the system with gene duplication produces a more accurate response (i.e. a response closer to the deterministic one), highlighting the role of gene copy number in noise buffering (*Rodrigo and Poyatos, 2016*).

In addition, we calculated the proposed fitness function for each single-cell response. Small gene expression inaccuracies (e.g. an excess of enzyme for the available substrate) can be perceived as a consequence of a hill-like fitness landscape in terms of the genotype-environment interaction (*Figure 1d*). To properly compare how each system of study resolves the fitness trade-off, we then calculated the selection coefficient for each response. We found a skewed distribution, peaked at 0 and with a positive mean of 0.08% (*Figure 3d*). This entails that phenotypic responses generated by duplicated genes give, on average, higher fitness values than responses generated by singleton genes. To better illustrate this fact, we represented cell fitness as a function of LacZ expression (*Figure 3e*), uncovering two reasons by which gene duplication is adaptive. In first place, the variance of the stochastic fluctuations (noise) in gene expression is reduced by 50% upon duplication (*Wang and Zhang, 2011*); when only intrinsic fluctuations are considered. However, when both intrinsic and extrinsic fluctuations are considered, the variance is reduced by 15–25%. In any case, this increases fitness on average, because the system displays a near-optimal behavior in the deterministic regime, thus fluctuations are costly. In second place, the population response upon

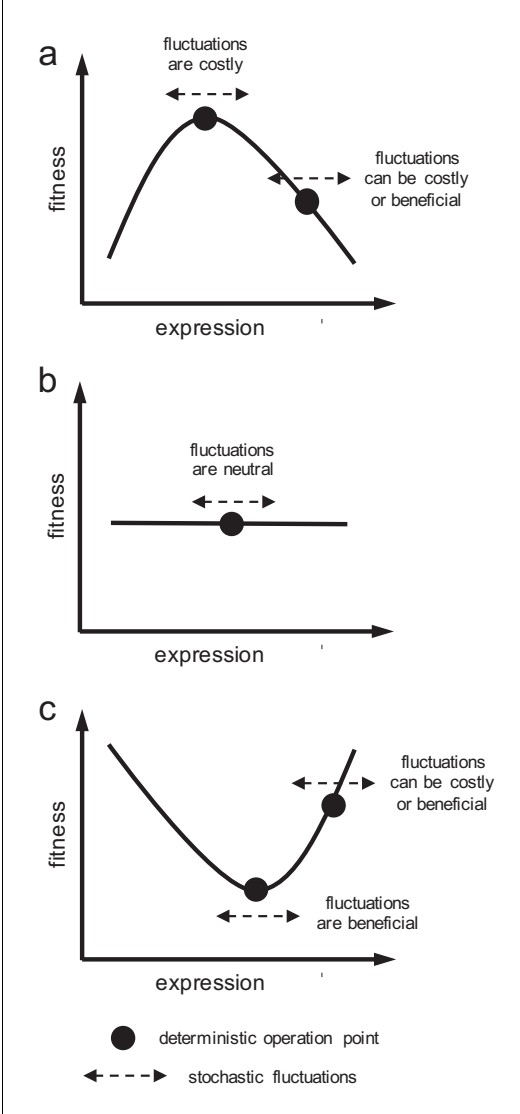

**Figure 2.** Schematics of cell fitness as a function of gene expression. Fitness function can (**a**) present a maximum, (**b**) be flat, or (**c**) present a minimum. Depending on the local shape, stochastic fluctuations in expression can be costly, beneficial, or neutral.
DOI: https://doi.org/10.7554/eLife.29739.003

duplication is slightly closer to the optimal operation point (*Figure 3e,f*). The model-based median dose-response curve (corresponding to the experimental response at the population level) is sigmoidal and has a Hill coefficient of 4 (*Dekel and Alon, 2005*). This results in a slope (LacZ vs. normalized lactose) of 1, calculated as $n/4$ at $x_0$ ($n$ is the Hill coefficient). This slope is higher than the slope coming from the optimal dose-response curve, which is 0.47 at $x_0$. However, when duplication is considered (maintaining the same expression levels), the median dose-response curve shows a slope of 0.75 (corresponding to an effective Hill coefficient of 3) also at $x_0$ (*Figure 3f*). This is because, in this case, the actual dose-response curve is more nonlinear than the optimal one, a feature that can indeed be amended by genetic redundancy (*Gammaitoni, 1995*; *Rodrigo and Poyatos, 2016*).

Finally, we calculated how much selection exists, on average, as a two-dimensional function of the magnitude of intrinsic noise and the concentration of lactose in the medium (*Figure 3g*). This highlights the fundamental link between noise reduction in gene expression and selective advantage (cell fitness). More in detail, we found that the higher the intrinsic noise, the higher the adaptive value of gene duplication. This is because intrinsic noise generates the required heterogeneity between the responses of the two gene copies to limit large stochastic fluctuations in the total gene expression. We also found that there is a maximal adaptive value of gene duplication at intermediate lactose doses, where the sensitivity of the system is the highest (*Figure 1f*). Out of this regime, the stochastic fluctuations, according to our simple mathematical model, have less impact on the phenotype (*Blake et al., 2006*).

## Gene duplication can be positively selected in a population thanks to more accurate responses

If gene duplication enhances cell fitness on average, viz., by reducing gene expression inaccuracies, it would be expected a positive selection of this trait in a population (*Kimura, 1983*). To verify this assumption, we performed experiments of in silico evolution (see Materials and methods), where a mixed population of cells carrying singletons and duplicates was monitored, considering equal LacZ expression in both types of cells (*Figure 4a*). The population was left to evolve without introducing any bias, with time-dependent stochastic fluctuations in gene expression uncorrelated from cell to cell. For simplicity, we simulated a scenario of experimental evolution (*Elena and Lenski, 2003*; *Dekel and Alon, 2005*), although the dynamics in nature might be more complex. We found that the frequency of cells carrying duplicates in the population increases with time, and that such an increase is well predicted by population genetic dynamics with the mean selection coefficient (*Figure 4b*). Notably, this points out that this parameter, which can be mathematically calculated

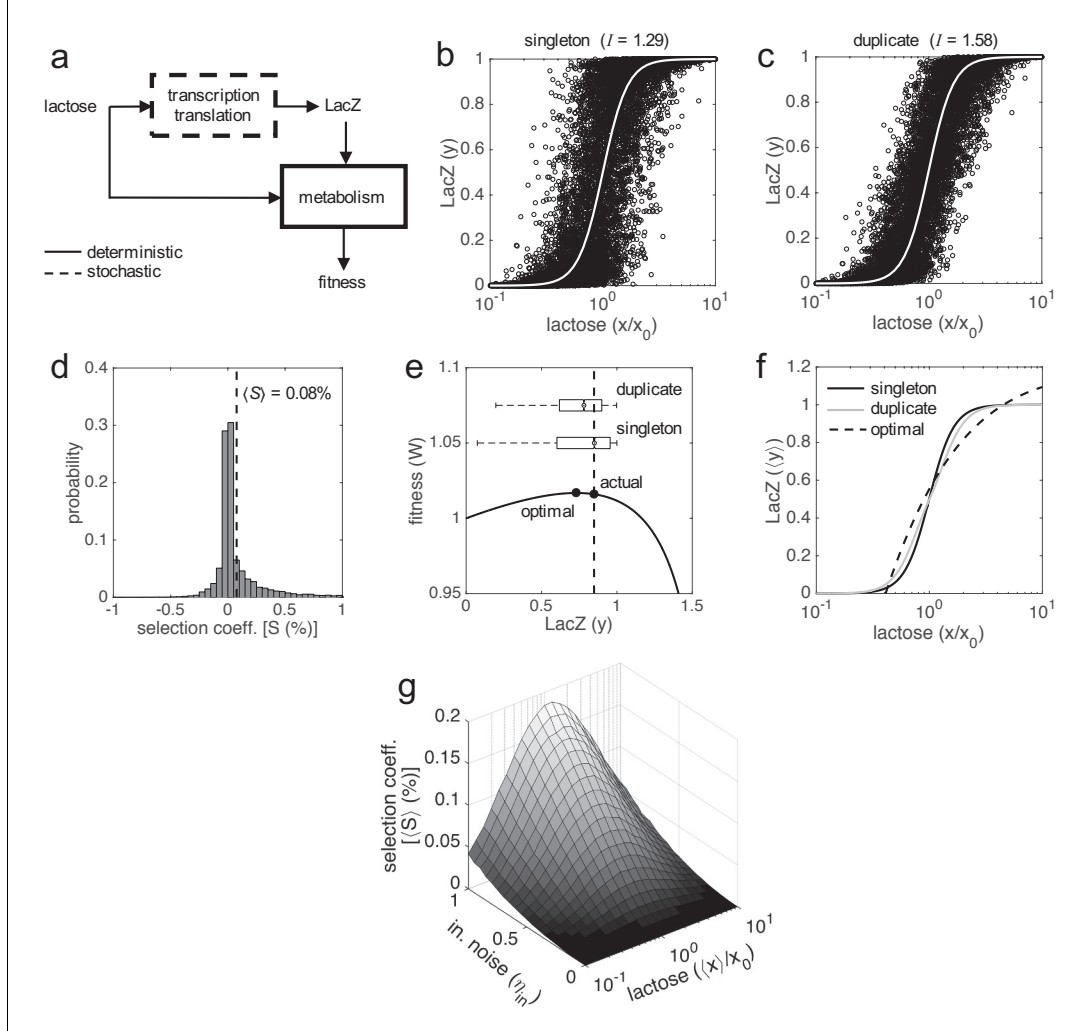

**Figure 3.** Selective advantage of gene duplication. (a) Block diagram of the system. Gene expression is calculated by means of a stochastic function, whilst fitness by means of a deterministic one. (b, c) Single-cell responses at different lactose doses (stochastic simulations, noise amplitudes of $\eta_{in}$ = 0.5 and $\eta_{ex}$ = 0). Lactose and LacZ concentrations are denoted by $x$ and $y$, respectively. The solid white line corresponds to the deterministic simulation. In b) the genotype contains a single copy of *lacZ* gene, whilst in c) it contains two copies. The value of mutual information ($I$) is shown in both cases: 1.29 bits of information in case of a singleton and 1.58 bits in case of a duplicate (about 25% increase in fidelity, significance assessed by a z-test, p ≈ 0 with $10^4$ points). (d) Selection coefficient ($S$) of a genotype with two copies of *lacZ* gene over another with just one copy. The mean selection coefficient is shown (dashed line). Skewness coefficient of 2.63. $W$ values calculated from $x$, $y$ values shown in b, c). (e) Fitness ($W$) as a function of LacZ (constant $x$ = 0.2 mM), showing the distributions of expression (boxplots) in case of one or two gene copies. The actual LacZ expression is shown (dashed line). (f) Dose-response curve between lactose concentration and the median LacZ expression ($\langle y \rangle$). The solid lines correspond to the actual responses in case of one (black) or two (gray) gene copies ($\eta_{in}$ = 0.5 and $\eta_{ex}$ = 0), whilst the dashed line corresponds to the optimal response. (g) Mean selection coefficient ($\langle S \rangle$) landscape of gene duplication as a function of the median lactose dose ($\langle x \rangle$, fluctuating dose) and the amplitude of intrinsic noise ($\eta_{in}$, with fixed $\eta_{ex}$ = 0.3). In all these plots, the expression levels of the duplicates with respect to the singletons are equal ($y_{max,1} = y_{max,2} = 0.5$).

DOI: https://doi.org/10.7554/eLife.29739.004

a priori, is sufficient to capture all the complexity underlying the stochastic evolutionary dynamics of the system (*Hegreness et al., 2006*).

In addition, we studied the effect of the magnitude of molecular noise. We distinguished between intrinsic and extrinsic noise (*Elowitz et al., 2002*). As predicted from our previous results, we found that the higher the intrinsic noise of the system, the higher the frequency of gene duplication in the population (*Figure 4c*). By contrast, the higher the extrinsic noise, the lower the frequency (*Figure 4c*), as this type of noise affects in the same way the responses of the two copies. Note that there is no gain following duplication when only extrinsic noise is considered. Furthermore, we

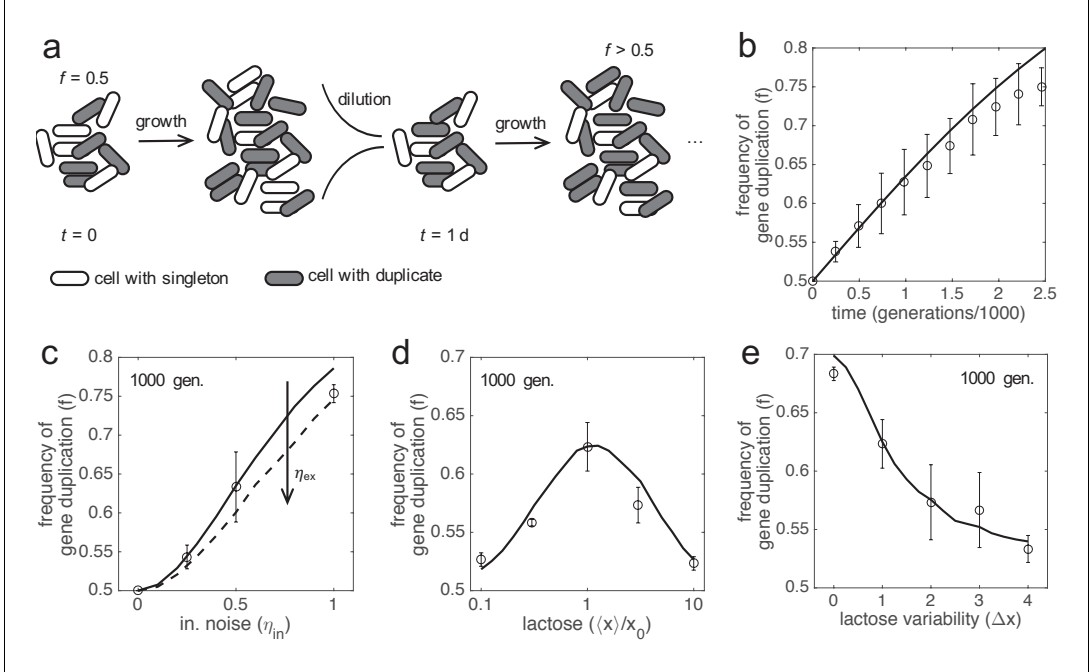

**Figure 4.** In silico evolution experiments. (**a**) Scheme of an evolutionary procedure, where serial dilution passages are applied, to assess the performance in a cell population of a genotype with two copies of *lacZ* gene over another with just one copy. (**b**) Time-dependent frequency of cells with gene duplication (*f*). Open circles and error bars correspond to experiments of in silico evolution (mean and standard deviation of three replicates) with an initial frequency of $f_0 = 0.5$, fluctuating lactose dose, and noise levels of $\eta_{in} = 0.5$ and $\eta_{ex} = 0$. The solid line corresponds to the theoretical prediction. (**c**) *f* at 1000 generations ($f_{1000}$) as a function of the amplitude of intrinsic noise ($\eta_{in}$). Experiments and prediction with $f_0 = 0.5$ and $\eta_{ex} = 0$. The dashed line corresponds to the theoretical prediction with $\eta_{ex} = 1$. (**d**) $f_{1000}$ as a function of the median lactose dose ($\langle x \rangle$). Experiments and prediction with $f_0 = 0.5$, $\eta_{in} = 0.5$ and $\eta_{ex} = 0.5$. (**e**) $f_{1000}$ as a function of the lactose fluctuation amplitude ($\Delta x$). $\Delta x = 0$ corresponds to constant lactose dose. Experiments and prediction with the same values of $f_0$, $\eta_{in}$ and $\eta_{ex}$ as in d). Three replicates were also considered in c, d, e). In all these plots, the expression levels of the duplicates with respect to the singletons are equal ($y_{max,1} = y_{max,2} = 0.5$).

DOI: https://doi.org/10.7554/eLife.29739.005

studied the effect of the environment (lactose dose). As predicted, we found an intermediate median dose at which the frequency of gene duplication in the population is the highest (*Figure 4d*). We also found that the higher the variance, the lower the frequency (*Figure 4e*). This is because, when lactose fluctuates from very low to very high doses, the signal-to-noise ratio is large enough to warrant a relatively accurate response with just one gene copy (*Hansen et al., 2015*). Of relevance, the population genetic dynamics in all these cases, with the corresponding mean selection coefficients, correctly explained the reported frequencies.

## Fixation is conditioned by the unexpected recurrence of formation and deletion of duplicates in a population

Gene duplicates can be spontaneously produced, through different mechanisms (*Hastings et al., 2009*), at very high rates in the cell. These rates, measured from experiments of mutation accumulation, go from $10^{-4}$ dup./gene/gen. in prokaryotes (*Reams et al., 2010*) to $10^{-7}$ dup./gene/gen. in higher eukaryotes (*Schrider et al., 2013*). Once produced, most of these duplicates are deleted as they are unstable, with a rate that appears to be higher than the formation rate (*Reams et al., 2010*; *Schrider et al., 2013*). In the particular case of the *lacZ* gene, we have a formation rate of $3 \cdot 10^{-4}$ dup./gene/gen. and a deletion rate of $4.4 \cdot 10^{-2}$ -/gene/gen. (in a single bacterial cell; data for *Salmonella enterica*). Therefore, gene duplication can be understood as a recurrent process that reaches an equilibrium point given by the ratio between the formation and deletion rates (*Figure 5a*), neglecting fitness effects. This equilibrium point would be lower if fitness effects (mostly detrimental) were considered. This entails about $2 \cdot 10^5$ cells carrying *lacZ* duplicates in a typical *E. coli* population of $2 \cdot 10^8$ cells in nature (*Lynch et al., 2016*; that is, frequency of about 0.1%). This surprising scenario

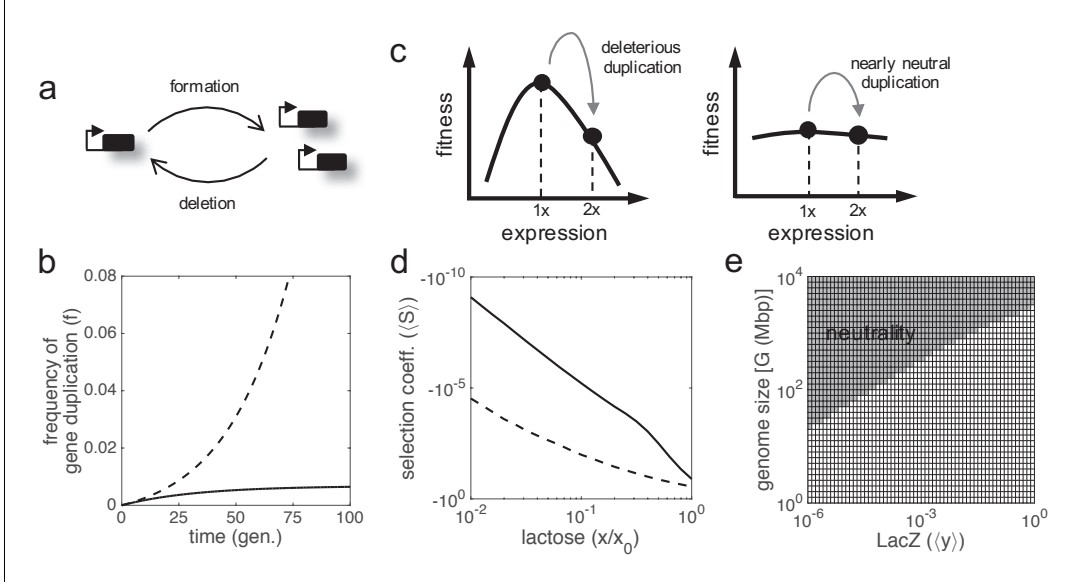

**Figure 5.** Gene duplication leading to double expression. (a) Scheme of the formation-deletion balance in gene duplication. (b) Time-dependent frequency of cells with gene duplication (f) when the formation and deletion rates of a second *lacZ* copy are considered. Sequence remodeling was not taken into account. The solid line corresponds to a scenario of neutrality, whilst the dashed line corresponds to a scenario of positive selection (with $S = 10\%$). (c) Schematics of fitness as a function of expression showing the effect of gene duplication. Two scenarios are considered: deleterious duplication (left; hill-like fitness landscape) and nearly neutral duplication (right; quasi-flat fitness landscape). (d) Mean selection coefficient ($\langle S \rangle$) as a function of lactose dose upon *lacZ* duplication doubling gene expression ($y_{max,1} = y_{max,2} = 1$). The solid line corresponds to noise levels of $\eta_{in} = \eta_{ex} = 0.3$ (moderate), whilst the dashed line corresponds to $\eta_{in} = \eta_{ex} = 1$ (high). (e) Identification of effectively neutral selective conditions (when $|\langle N \rangle \cdot \langle S \rangle| < 1$, region shaded) in terms of gene expression ($y$) and genome size ($G$), which determines the effective population size ($\langle N \rangle$). In this context, no benefit was considered ($a = 0$), with moderate noise levels.

DOI: https://doi.org/10.7554/eLife.29739.006

has an immediate consequence, viz., duplicated genes cannot be fixed in the population by drift under neutral selective conditions (*Figure 5b*); a result already anticipated (*Clark, 1994*) in clear discrepancy with the conventional wisdom (*Lynch, 2007*). Indeed, the formation-deletion balance would always take the system to the same equilibrium point.

However, the preceding argument only focuses on a static picture, ignoring the dynamics of the genetic process. In bacteria (*lacZ* gene), the time to reach the equilibrium point is about 68 generations (three times the inverse of the deletion rate), which is a relatively short transient period. By contrast, in flies (*Drosophila melanogaster*), the formation rate is of $10^{-7}$ dup./gene/gen. and the deletion rate of $10^{-6}$ -/gene/gen. (*Schrider et al., 2013*). Although this would yield equilibrium frequencies up to 10%, the transient periods would be longer than $10^6$ generations (0.2 Ma in natural conditions; *Pool, 2015*). Fixation could then happen by drift, as their effective population sizes are of $10^6$ flies (*Lynch et al., 2016*), although not persistently. Note that the inverse of this number indeed specifies an upper limit for the deletion rate. In addition, the formation-deletion balance could be shifted if further genome rearrangements affecting duplicated genes were considered, such as gene relocation (about $10^{-11}$ fixed rearr./gene/gen. for *D. melanogaster*; *Ranz et al., 2001*). In effective terms, gene relocation would reduce the deletion rate, and, consequently, fixation would be more likely (*Wong and Wolfe, 2005*). Such a relocation would also shift the intrinsic-extrinsic noise balance toward more uncoupled responses (*Becskei et al., 2005*), which could enhance the benefit by intrinsic noise reduction.

## Most of the new-born duplicates lead to increased expression and are costly for the cell

So far, we have demonstrated that a duplicated gene offers a selective advantage provided the total gene expression level is maintained, with one or two copies (gene dosage sharing). However, this condition is not usually met during the critical very initial phase, when the duplicate has just born. In

general, we can assume that the expression level is doubled upon duplication, although this may vary due to the particular position in the chromosome of the duplicated gene and the type of cell (*Stranger et al., 2007*). Certainly, an increase of expression due to gene duplication is detrimental in most environments (*Figure 5c,d*; *Price and Arkin, 2016*), thus positive or neutral selective conditions are difficult to invoke to explain the fixation of these type of genotypic changes, mainly in prokaryotes and lower eukaryotes (*Lynch and Marinov, 2015*). For instance, at constant 0.13 mM lactose, we obtained mean selection coefficients between $-28\%$ (at very high noise levels) and $-1\%$ (at no noise) upon duplication of the *lacZ* gene (assuming double expression), which yield negligible fixation probabilities (almost 0) for a sufficiently large bacterial population. It can be argued, nevertheless, that the cost of over-expression decreases as long as the genome size increases (*Lynch and Marinov, 2015*). This assumption, together with the negative correlation between complexity and population size (*Lynch and Conery, 2003*), makes effectively neutral selective conditions plausible to rationalize the fixation of duplicates that are expressed (e.g., essential genes) in higher eukaryotes (*Figure 5e*; *Makino et al., 2009*).

Only in absence of lactose, when the enzyme is not needed, the duplication is strictly neutral (no benefit, no cost due to regulation). But neutral selective conditions can be reached *de facto* if the absolute value of the selection coefficient is lower than the inverse of the effective population size (*Kimura, 1983*). This condition is challenging for prokaryotes, as their population sizes are very large (*Lynch and Conery, 2003*). In our particular case, we obtained mean selection coefficients in the order of $-10^{-10}$ (at moderate noise levels) when the nutrient amount is scarce (1 µM lactose), which could favor the fixation of a *lacZ* duplicate by genetic drift.

## Gene dosage sharing upon duplication, fitness increase on average, and estimation of the fixation probability

Can a cell carrying a new-born duplicate that is expressed (in principle, in an operation point close to a local optimum) overcome the cost of an additional copy and then invade the population without invoking the need for more expression (to face an extreme environment)? We here predicted that the genetic variability existing in a population would allow reaching adaptive gene duplications (*Figure 6a*). Mutations in the *cis*-regulatory region of the *lacZ* gene may change its wild-type expression level. According to previous results (*Otwinowski and Nemenman, 2013*), the distribution of mutations in terms of maximal promoter activity is peaked at 1, but skewed to the left (*Figure 6b*). This indicates that about 10% of them yield cells with nearby 50% lower expression. Thus, if a gene duplication event occurred in one of these cells, the genotypic change would be selectively advantageous (*Figure 6c*). The frequency of such cells in the population depends, of course, on the mutation rate; the greater the ability to generate genetic diversity, the higher the chances to reach adaptive duplications. For *E. coli*, where the per base mutation rate is of $10^{-10}$ mut./bp/gen. (*Lee et al., 2012*), this frequency can be estimated in $10^{-9}$ (i.e. 0.2 mutants with nearby 50% lower expression per generation in a natural population of $2 \cdot 10^{8}$ cells). Hence, the probability that a duplication and such a mutation concur in the same cell in a generation (duplication after promoter mutation) is of $10^{-4}$ ($=0.2 \cdot 10^{-3}/2$; i.e., 1 suitable concurrence each $10^{4}$ generations).

In particular, at constant 0.13 mM lactose, we obtained a relatively high mean selection coefficient of 0.19% when the wild-type expression is recovered upon duplication (in a highly noisy scenario). However, the selection coefficient has to be greater than the duplication deletion rate to ensure fixation (*Figure 5b*); a condition that is not met here. Certainly, the high deletion rates observed in bacteria (*Reams et al., 2010*) protect them from acquiring genetic redundancy (perhaps, this is why *lacZ* is not duplicated in *E. coli* despite this may be beneficial). In other local genetic contexts, also in bacteria, the deletion rate of a *lacZ* duplicate can be as low as $4.1 \cdot 10^{-4}$ -/gene/gen. (*Reams et al., 2012*). In this scenario, a selection coefficient of 0.19% would lead to fixation. We then estimated a global fixation probability of $3 \cdot 10^{-7}$ (= $2.15 \cdot 10^{-4} \cdot 10^{-4}$; *Figure 6d*; see Materials and methods). Remarkably, our estimation is much higher than $5 \cdot 10^{-9}$, the fixation probability under hypothetical neutrality (*Kimura, 1983*).

A fitness increase on average due to expression noise reduction could also lead to the fixation of duplicates in eukaryotes, as nothing prevents assuming the same positive selective conditions (*Raser et al., 2004*; *Hansen et al., 2015*), which now largely outperform the duplication deletion processes. For *D. melanogaster*, for instance, where the per base mutation rate is of $5 \cdot 10^{-9}$ mut./bp/gen. (*Schrider et al., 2013*), and complete gene duplications have little impact on fitness

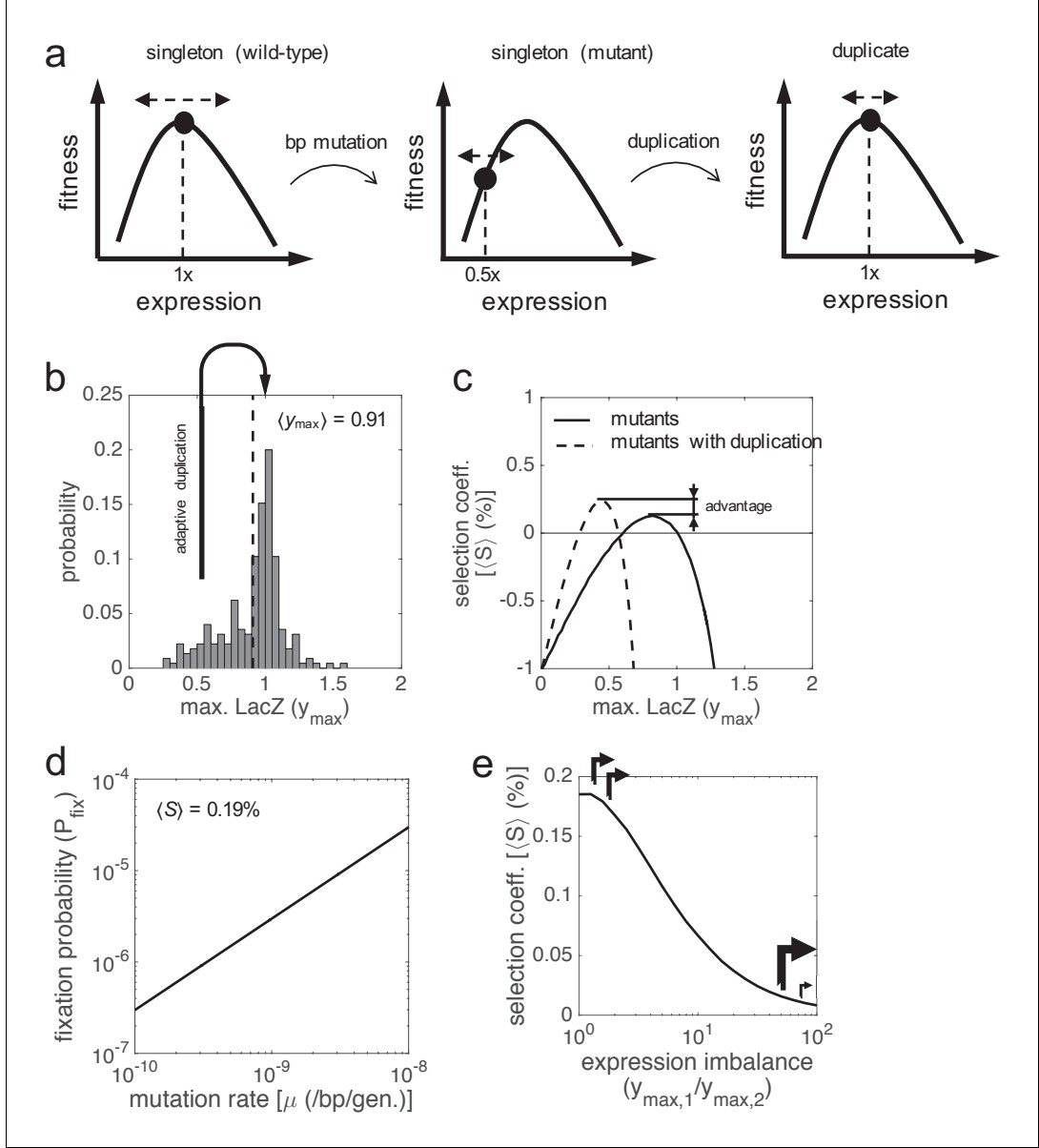

**Figure 6.** Gene duplication leading to maintained expression. **(a)** Schematics of fitness as a function of expression showing a path to reach adaptive gene duplications without the need for more expression. Two steps are considered: first a base-pair mutation that reduces in half the expression level, and then a duplication that recovers the ancestral level. **(b)** Distribution of the activity of *lac* promoter mutants based on experimental data, as the maximal LacZ expression ($y_{max}$, irrespective of lactose dose). The mean activity is shown (dashed line). Skewness coefficient of −0.68. **(c)** $\langle S \rangle$ of the promoter mutants versus the wild-type system (solid line), with fluctuating lactose dose and high noise levels. The dashed line corresponds to the comparative between promoter mutants that duplicated the *lacZ* gene and the wild-type system. **(d)** Fixation probability ($P_{fix}$) of gene duplication as a function of the mutation rate of the cell (μ), with $\langle S \rangle$ = 0.19% and $\langle N \rangle$ = 2·10⁸. **(e)** $\langle S \rangle$ as a function of the expression imbalance between the two *lacZ* copies ($y_{max,1}$ / $y_{max,2}$), when the system recovers its ancestral expression levels ($y_{max,1}$ = $y_{max,2}$ = 0.5), with constant $x$ = 0.13 mM and high noise levels. Arrows illustrate the corresponding promoter strengths.

DOI: https://doi.org/10.7554/eLife.29739.007

(*Emerson et al., 2008*; note that other genome rearrangements not affecting entire genes are significantly deleterious), we estimated that 0.05 mutants with nearby 50% lower expression and up to $10^5$ duplicants of the gene of interest would be found in the natural population. Hence, the probability of concurrence in the same organism (duplication after promoter mutation) would be of $2.5 \cdot 10^{-3}$. Consequently, the global fixation probability would be of $10^{-5}$; again, higher than the one under hypothetical neutrality (*Kimura, 1983*).

## Maintenance of a duplicate upon fixation in the population

A forthcoming change in lactose dose would be highly detrimental if a second *lacZ* copy were fixed in the population either under neutrality due to insignificant expression or under strong selection due to expression demand. In the former case, an increase of lactose would be detrimental; in the latter, a decrease would. Consequently, either the elimination of the duplicate by purifying selection (*Lynch and Conery, 2000*) or the accumulation of mutations that lower the LacZ expression to recover the ancestral phenotype (*Force et al., 1999*; *Qian et al., 2010*) would be promoted; with clonal interference in the case of asexual populations (*Rozen et al., 2002*; *Desai et al., 2007*). In the latter case, the two gene copies could be maintained in the genome for long time by buffering of costly stochastic fluctuations of intrinsic nature if they held similar expression levels (*Figure 6e*; *Gout and Lynch, 2015*); otherwise the gain in accuracy decreased. Conversely, if a second *lacZ* copy were fixed according to the path shown in *Figure 6a* under weak selection, it would be safe from changes in lactose dose.

The genomic inspection of organisms in which genetic drift is not, in principle, a suitable force to drive the fixation of duplicates (e.g. bacteria or yeast; *Lynch and Marinov, 2015*) gave us some empirical insight, despite the masking produced by subsequent evolutionary trajectories. In many cases of duplication, there is no a significant increase in total expression (e.g. duplicates in *Saccharomyces cerevisiae* vs. singletons in *Schizosaccharomyces pombe*; *Qian et al., 2010*). Thus, either duplicates were fixed by dosage in a definite environment to then return to ancestral expression levels, or duplicates were fixed by other means. In any case, the preservation of the ancestral function in the second copy is expected (*DeLuna et al., 2008*). Whether noise reduction was actually relevant for some fixations or not is hard to say without conducting an experimental approach to measure variability and selection (revealing the fitness landscape; *Figure 2*); notwithstanding, it seems a plausible mechanism according to our results, already put forward with the computational analysis of gene expression patterns (*Lehner, 2010*) and metabolic flux balances (*Wang and Zhang, 2011*) in yeast.

If dosage mattered at some point, the function encoded by the duplicated gene would be more important at the time of duplication than today. In *E. coli*, for example, genes *fsaA* and *fsaB* are paralogs, with high sequence (69%) and functional similarity, coding for a genuine fructose-6-phosphate aldolase (*Sánchez-Moreno et al., 2012*). The relevance of this enzyme for today *E. coli* is unclear, suggesting that *fsaB* might have been fixed by dosage in past habitats in which rare sugars were frequent. However, if noise were the critical aspect, the system would present some regulation to link environment with phenotype and the function would be of routine for the cell. In particular, *E. coli* expresses two redundant gluconokinases, encoded in genes *gntK* and *idnK* (51% of sequence identity), to face environments in which gluconate is the carbon source due to glucose oxidation (*Vivas et al., 1994*). Similar to the regulation of *lacZ* by lactose (*Jacob and Monod, 1961*), gluconate activates the expression of *gntK* and *idnK* by inhibiting the transcriptional repressor GntR (*Afroz et al., 2014*). Again, there would be a trade-off between metabolic benefit and expression cost (*Figure 1c*; read gluconate instead of lactose and GntK/IdnK instead of LacZ). Arguably, duplication might have been fixed in this case to cope with gene expression inaccuracies, especially when GntR produces bimodal responses (captured in single-cell experiments; *Afroz et al., 2014*).

## A comprehensive model compatible with population genetics to explain the early fate of gene duplications

Taking all our results together, we formulated a comprehensive model to explain the early fate (viz., fixation or elimination) of gene duplications (*Figure 7*). Notably, this model is compatible with population genetics, involving positive and neutral selective conditions (*Lynch, 2007*). On the one hand, a significant number of duplicates could be fixed by genetic drift only in complex organisms (i.e.

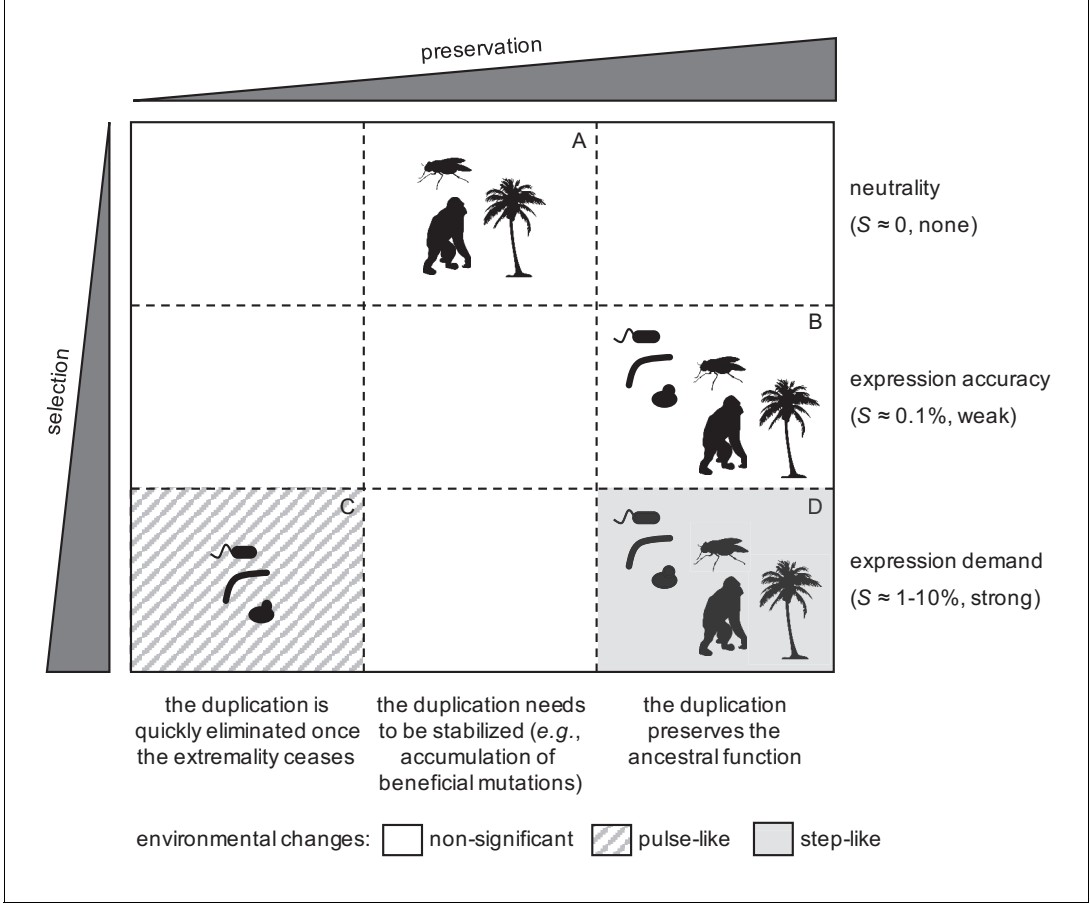

**Figure 7.** General model to explain the fixation of duplicated genes as a function of the degree of selection in the population and preservation in the genome for long time. Representative silhouettes correspond to bacteria (prokaryotes), yeasts (lower eukaryotes), insects, plants, and mammals (higher eukaryotes).

DOI: https://doi.org/10.7554/eLife.29739.008

higher eukaryotes; sector A in *Figure 7*). This would be due to their increased ability to allocate additional resources for expression (*Lynch and Marinov, 2015*), and their apparently reduced duplication deletion rate with respect to the inverse of the population size (*Schrider et al., 2013*). However, these fixed duplications would not be stable, due to the formation-deletion balance (*Reams et al., 2010*), and then, for a long-term preservation, they would require the accumulation of beneficial mutations (*Han et al., 2009*), or the relocation of the second copy in the genome to prevent its deletion (*Ranz et al., 2001*). This would lead to late fates of sub- or neo-functionalization (*Force et al., 1999*; *Conant and Wolfe, 2008*).

On the other hand, positive selection could drive the fixation of duplicates in both complex and simple organisms. When the environmental changes were relatively rapid, only organisms with short generation times (i.e., prokaryotes and lower eukaryotes) could fix duplications (sector C in *Figure 7*; *Riehle et al., 2001*). However, such duplications would be quickly eliminated from the population afterwards (once the environment changed again), as the genome rearrangement rates are orders of magnitude higher than the per base mutation rates (*Reams et al., 2010*). By contrast, when a given environmental change were prolonged, any organism, irrespective of its generation time, could fix duplications (sector D in *Figure 7*; *Emerson et al., 2008*). In this case, they would be under strong positive selection, and, consequently, they would be preserved for long time. Furthermore, all organisms could fix duplications by producing more accurate responses (sector B in *Figure 7*), without the need of significant environmental changes; provided the gene of interest were noisily expressed (*Elowitz et al., 2002*; *Raser et al., 2004*), and the duplication deletion rate were lower

than the weak selective advantage. In the very long term, these weak selective conditions could also allow the exploration of novel functions, as they ensure the preservation of duplicates, without invoking fortuitous exploration in the ancestral state (*Bergthorsson et al., 2007*), and with amplification when the advantage provided by the narrowed novel function were higher than the advantage by noise reduction.

## Discussion

The inherently stochastic nature of gene expression is certainly an evolutionary driver when it is linked to cell fitness to dictate the selection of particular genetic architectures (*Batada and Hurst, 2007*; *Maamar et al., 2007*). Our results demonstrate that gene duplication can be positively selected as an architecture that allows enhancing information transfer in genetic networks (i.e. mitigation of expression errors; *Rodrigo and Poyatos, 2016*). Accordingly, the genetic robustness indeed observed upon the accumulation of genetic redundancy (*Keane et al., 2014*) would be more a consequence than a selective trait (*Kafri et al., 2006*). Certainly, by aggregating the responses of two genes, intrinsic fluctuations can be mitigated, but not fluctuations of extrinsic nature. This way, duplication would be more favorable in scenarios in which intrinsic noise is preponderant. The balance between intrinsic and extrinsic noise depends on the particular environmental conditions and the regulatory structures in which the gene in embedded. Intrinsic noise can be significant when the medium is rich in nutrients, the expression levels are low, and no further regulations affect the gene (*Swain et al., 2002*). For example, competence in *Bacillus* is mainly governed by intrinsic noise (*Maamar et al., 2007*). To follow our model, noise has to mainly impinge the regulation of the system, that is, disturb the link between the signal molecule and gene expression (*Blake et al., 2006*). Moreover, our results highlight that a population genetic model with the mean selection coefficient is enough to explain the complex, stochastic evolutionary dynamics of duplication fixation. Of note, the reported intrinsic adaptive value, which cannot be captured by sequence analyses, was derived from basic mathematical models of gene regulation and cell fitness (*Dekel and Alon, 2005*).

Notably, we anticipated a series of testable results by following our theory of error buffering upon duplication. First, the gene expression level is indicative of the fixation path. The theory requires that gene expression is roughly maintained (i.e. gene dosage sharing, duplicates vs.. singletons), with the aim of minimizing deleterious fitness effects. This would hold for several fixed duplicates in different organisms (*Qian et al., 2010*; *Gout and Lynch, 2015*; *Cardoso-Moreira et al., 2016*; *Lan and Pritchard, 2016*), although most of the formed duplicates would be under strong purifying selection due to the cost of over-expression, as already proposed (*Lynch and Conery, 2000*). By contrast, those fixed duplicates showing increased gene expression levels would reflect the effect of genetic drift (*Lynch and Conery, 2003*) or positive selection for dosage after prolonged environmental changes (e.g. the case of flies; *Emerson et al., 2008*; *Cardoso-Moreira et al., 2016*).

Second, noisy genes are expected to be more duplicable (e.g. as it seems to happen in yeast; *Lehner (2010)*; *Dong et al., 2011*) when noise has deleterious fitness effects. Indeed, the gain experimented by the system upon duplication is greater when gene expression inaccuracies are significant (*Rodrigo and Poyatos, 2016*). This would explain the TATA box enrichment in the *cis*-regulatory regions of duplicated genes, as these genetic motifs are associated to high plasticity (i.e. high sensitivity to multiple environmental changes) and high gene expression noise by inducing transcriptional bursts (*Blake et al., 2006*; *Lehner, 2010*). Note that if noise were beneficial (e.g. as a survival strategy in fluctuating environments; *Acar et al., 2008*), duplication would not be favored. Moreover, we might argue that essential genes would be less duplicable (*He and Zhang, 2006*) as a consequence of their reduced gene expression noise (*Batada and Hurst, 2007*). Genes under the control of regulatory structures that buffer noise (e.g. negative feedbacks) would not be duplicable either (*Warnecke et al., 2009*). However, this consideration should be taken with caution, as genes not essential a priori could be duplicated and then, upon fixation, accumulate beneficial mutations (*Han et al., 2009*) to ensure preservation for long time, resulting a posteriori in essential genes due to functional diversification (as it seems in the case of mammals; *Makino et al., 2009*).

Third, the local genetic context would be highly determinant of the fixation of a duplicate (*Reams et al., 2012*), explaining why some genes are more duplicable than others in scenarios of apparent neutrality (hot spots; *Perry et al., 2006*). Moreover, duplicates would be much shorter lived in prokaryotes than in eukaryotes (*Lynch and Conery, 2003*), due to the differences of orders

of magnitude in the duplication deletion rates. After all, the precise experimental determination of the molecular rates of gene copy number variation would unveil to what extent natural selection has actually rivaled random genetic drift to shape complexity along the course of life history (*Rodrigo, 2017*).

These predictions involve, nevertheless, some limitations. On the one hand, due to a simplified mathematical model not considering the many molecular/genetic attributes that impinge implicitly on gene expression, such as promoter sequence-dependent noise levels (*Metzger et al., 2015*), response coupling due to genetic proximity (*Becskei et al., 2005*), or recursive fitness-expression dependence (*Klumpp et al., 2009*). On the other hand, due to the difficulty to provide direct empirical evidence supporting the fixation of duplicates by reducing intrinsic noise. In this regard, we expect to carry out in the future an experimental approach (*Dekel and Alon, 2005*; *Keane et al., 2014*) complementary to this theoretical study. Despite these edges, our results complete a mechanistic model in which duplicates are fixed either by genetic drift (no selection) or by gene dosage (strong selection) with the addition of a new principle, viz., reduction of gene expression inaccuracies upon duplication can result in a weak selective advantage.

## Materials and methods

### Fitness function

The *lac* operon of *E. coli* (*Jacob and Monod, 1961*) was considered as a biological model system from which to apply a mathematical framework, and cell growth rate was taken as a metric of fitness (*W*; *Elena and Lenski, 2003*). In this particular case, the benefit function reads $B = a \cdot y \cdot x / (k + x)$, where $a$ accounts for the increase in growth rate due to lactose utilization ($x$ denotes its concentration; $y$ denotes the normalized LacZ expression), and $k$ is the Michaelis-Menten constant. In addition, the cost function reads $C = b \cdot y / (h - y)$, where $b$ accounts for the decrease in growth rate due to LacZ expression, and $h$ for the maximal resources available in the cell (*Dekel and Alon, 2005*). Thus, the fitness function reads $W = W_0 \cdot (1 + B - C)$, where $W_0$ is the cell growth rate in absence of lactose ($x = 0$). Note that this model underestimates the adaptive ability of the bacterium by not considering the effect of LacY. Moreover, the normalized LacZ expression, in the deterministic regime, is given by $y = x^n/(x_0^n + x^n)$, where $x_0$ is the lactose regulatory constant, and $n$ the Hill coefficient (accounting for the regulatory sensitivity). In this model, LacZ is not expressed in absence of lactose. If $y > h$, we assumed $W = 0$. All parameter values were experimentally fitted, resulting in $W_0 = 1$ h$^{-1}$, $a = 0.17$, $k = 0.40$ mM, $b = 0.036$, $h = 1.80$, $x_0 = 0.13$ mM, and $n = 4$ (*Dekel and Alon, 2005*). The optimal LacZ expression ($y_{opt}$) was obtained by imposing $dW/dy = 0$, resulting in $y_{opt} = h - [b \cdot h \cdot (k + x) / (a \cdot x)]^{1/2}$.

### Stochastic gene expression

The normalized LacZ expression in presence of molecular noise was modeled, in steady state, as $y = y_{max} \cdot (x \cdot z_1 \cdot z_0)^n / [x_0^n + (x \cdot z_1 \cdot z_0)^n]$, where $y_{max}$ is the maximal expression level (in general, $y_{max} = 1$), and $z_1$ and $z_0$ random variables accounting for intrinsic and extrinsic noise sources, respectively. Here, they were log-normally distributed [with mean 0 for both $\log(z_1)$ and $\log(z_0)$, and standard deviation $\eta_{in}$ for $\log(z_1)$ and $\eta_{ex}$ for $\log(z_0)$]. This accounts for the noisy de-repression of the promoter and subsequent expression due to lactose. Note that whilst LacZ can show a bistable expression pattern with non-metabolizable synthetic compounds (*Ozbudak et al., 2004*), its expression is monostable with lactose (*van Hoek and Hogeweg, 2006*). For simplicity, the transient LacZ expression was overlooked, and the noise levels were considered constant during a cell cycle. The median response of a population is denoted by $\langle y \rangle$.

Typical values characterizing the magnitude of the stochastic fluctuations ($\eta_{in}$ and $\eta_{ex}$) range between 0.1 and 0.5. They lead to values of gene expression noise (understood as the coefficient of variation) between 0.26 and 0.72 (in the case of $\eta_{in} = \eta_{ex}$ and $x = x_0$), in agreement with experimental reports (*Elowitz et al., 2002*).

### Gene duplication

The combined expression of two genes coding for LacZ in presence of molecular noise was modeled as $y = y_{max,1} \cdot (x \cdot z_1 \cdot z_0)^n / [x_0^n + (x \cdot z_1 \cdot z_0)^n] + y_{max,2} \cdot (x \cdot z_2 \cdot z_0)^n / [x_0^n + (x \cdot z_2 \cdot z_0)^n]$, where $z_2$ is a random

variable accounting for intrinsic noise on the second copy, with the same distribution as for $z_1$ ($z_1$ and $z_0$ as before). Note that whilst extrinsic fluctuations ($z_0$) are common, intrinsic fluctuations ($z_1$ and $z_2$) are independent for each gene copy (*Elowitz et al., 2002*). Moreover, the expression levels of the duplicates with respect to the singletons can be adjusted with the values of $y_{max,1}$ and $y_{max,2}$, with $y_{max,1} = y_{max,2} = 0.5$ for equal total expression, and $y_{max,1} = y_{max,2} = 1$ for double expression.

In addition, the bacterial model was modified to simulate the effect of gene duplication in organisms of different complexity. For that, the parameter $h$ in the cost function was set in terms of the genome size ($G$, in Mbp of haploid genome), simply as $h \approx 0.36 \cdot G$ (e.g. $G \approx 5$ for *E. coli*, or $G \approx 3000$ for *H. sapiens*), assuming that complex organisms have more resources to accommodate new gene expressions (*Lynch and Marinov, 2015*). The effective population size (here denoted by $\langle N \rangle$), determinant of the fixation of new genotypes, was also set in terms of $G$, resulting in $\langle N \rangle \approx 3 \cdot 10^9 / G^{1.44}$; an equation roughly inferred from previously reported estimates (*Lynch and Conery, 2003*).

## Information transfer

Mutual information ($I$) was used as a metric to characterize information transfer by considering the system as a communication channel between the environmental molecule (lactose) and the functional protein (enzyme, LacZ) resulting from gene expression. $I$ was calculated as previously done (*Rodrigo and Poyatos, 2016*), between $\log(x)$ and $y$. To model the variation of lactose, a random variable log-normally distributed was considered [with mean 0 and standard deviation 1, otherwise specified, for $\log(x/x_0)$]. The median lactose dose is denoted by $\langle x \rangle$, and the fluctuation amplitude, denoted by $\Delta x$, corresponds to the standard deviation of $\log(x)$. To compare statistically two $I$ values, we followed the approximation proposed by *Cellucci et al. (2005)* to obtain an equivalent correlation coefficient, and then the Fisher's $r$-to-$z$ transformation.

## Genotype-phenotype map

Here, the LacZ expression defines the phenotype of the cell (i.e. its metabolic capacity), and for the wild-type genotype it is lactose dependent through the LacI regulation (*Jacob and Monod, 1961*). Because differences in fitness are very small, the normalized expression ($y$) was assumed independent of it (*Klumpp et al., 2009*). Potential beneficial mutations are those that change the *lac* promoter activity (the *cis*-regulatory regulatory region of LacZ, of about $10^2$ bp). According to an analysis of a large library of mutants (*Kinney et al., 2010*) resulting in a linear model of categorical variables (*Otwinowski and Nemenman, 2013*), the distribution of maximal LacZ expression upon single-point mutations was inferred. For simplicity, no epistatic interactions were taken into account, although they could matter. Mutations were also assumed to affect only the mean expression level and not the noise, even though this latter might happen (*Metzger et al., 2015*).

## In silico evolution

A medium with maximal capacity for $N = 10^5$ cells was considered, and serial dilution passages were simulated (*Elena and Lenski, 2003*), with a dilution factor of $D = 100$ (in terms of volume, with deterministic dominance). The dilution period was set to 1 d. Lactose also varied with the same period. The doubling time of a given cell was $1/W$, with $W$ calculated from the stochastic LacZ expression. In case of no saturation, the cell volume increased as $2^{W \cdot t}$, where $t$ is the time in h. Because doublings occur in about 1 h, the number of generations per passage is bounded to $\log_2(D) = 6.64$. Two genotypes were put in competition: one with a single copy of LacZ, the other with two copies. No mutations were allowed to occur.

## Population genetics

In scenarios of competition between two subpopulations (i.e. two different genotypes), the ratio between them ($r$) reads $r = r_0 \cdot 2^{S \cdot t}$, where $r_0$ is the initial ratio, $S$ the selection coefficient, and $t$ the time measured in generations (*Hegreness et al., 2006*). By setting $W$ and $W'$ the fitness values of each genotype (with $W' > W$), the selection coefficient is calculated as $S = W'/W - 1$. When fitness changes over time, the mean selection coefficient ($\langle S \rangle$) is used. The frequency of the genotype with advantage in the population is $f = 1/(1 + 1/r)$. The dynamics of a punctual beneficial mutant appeared in an evolutionary experiment of serial dilution passages, with maximal population size $N$ and dilution factor $D$, is given by $r = 2^{S \cdot t} / \langle N \rangle$, where $\langle N \rangle = N / D^{1/2}$ is the geometric mean

population size (also considered the effective population size; *Lewontin and Cohen, 1969*). The fixation probability is $P_{fix} = 2S$, and the characteristic fixation time $t_{fix} = \log_2(\langle N \rangle^2)/S$. Note that the time for 50% invasion of the population is $t_{half-fix} = \log_2(\langle N \rangle)/S = t_{fix}/2$. However, we have $P_{fix} = 1/\langle N \rangle$ and $t_{fix} = 2\langle N \rangle$ for a neutral mutant (*Kimura, 1983*).

By contrast, if multiple beneficial mutants are recurrently produced at rate $\mu_b$, the dynamics is given by $r = \mu_b \cdot N \cdot 2^{S \cdot t} / [S \cdot \log(D) \cdot \langle N \rangle] \approx \mu_b \cdot 2^{S \cdot t} / S$, as in each passage $\mu_b \cdot N$ different mutants are generated (valid for $\mu_b \cdot N > 1$; *Desai et al., 2007*). Because mutants are now recurrent, $P_{fix} = 1$, and the characteristic fixation time reads $t_{fix} = \log_2[\langle N \rangle \cdot S / \mu_b]/S$. When $m$ different mutations accumulate successively, $t_{fix} \approx t_{fix}(m) + t_{half-fix}(m-1) + \ldots + t_{half-fix}(1)$, that is, a subsequent mutation can start its fixation when the preceding mutation has invaded the 50% of the population (*Lang et al., 2013*). If $\mu_b \cdot N \ll 1$, the system can be treated as in the case of a punctual beneficial mutation, and the dynamics can be written as $r = 2^{S \cdot (t - T)} / \langle N \rangle$, with a delay of $T = \log_2(D) / (\mu_b \cdot N)$, the mean number of generations required to produce a mutant, and $P_{fix} = 2S$.

Moreover, in case of gene duplication, if multiple beneficial mutants are recurrently produced at rate $\mu_c$, and deleted at rate $\mu_d$, the dynamics is given by $r \approx \mu_c \cdot 2^{S' \cdot t} / S'$, with $S' = S - \mu_d$ as an effective selection coefficient (valid for $\mu_c \cdot N > 1$, and $S > \mu_d$). Again, if $\mu_c \cdot N \ll 1$, the system can be treated as in the case of a punctual beneficial mutation, with $P_{fix} = 2S'$. If $S \ll \mu_d$, the stationary solution can be approached by $r \approx \mu_c / \mu_d$ for effectively neutral mutations, or by $r \approx \mu_c / (\mu_d - S)$ for deleterious mutations.

## Genetic diversity

The per base mutation rate of *E. coli* is $\mu = 10^{-10}$ mut./bp/gen. (*Lee et al., 2012*). Cultures of this bacterium may reach population sizes up to $N = 10^9$ cells ($\langle N \rangle = 2 \cdot 10^8$). This means, on average, 0.02 (= $\mu \cdot \langle N \rangle$) mutants of a given base pair in the population. The number of base pairs, mainly in the *cis*-regulatory regulatory region, whose mutation reduces in half the expression of a gene of interest can be estimated in 10 (based on data for *lacZ*). Thus, $\mu_b = 10 \cdot \mu$, which means 0.2 (= $\mu_b \cdot \langle N \rangle$) mutant of this type in the population on average. This frequency may even be higher if we not only consider the mutations in the *lac* promoter, but also the mutations in the coding region, or affecting the activity of its regulators (e.g. CRP; *Kinney et al., 2010*).

In addition, for the *lacZ* gene, its duplication formation rate is of $\mu_c = 3 \cdot 10^{-4}$ dup./gene/gen., and its duplication deletion rate of $\mu_d = 4.1 \cdot 10^{-4} – 4.4 \cdot 10^{-2}$ -/gene/gen. (*Reams et al., 2010*; *Reams et al., 2012*). In absence of lactose, duplications are neutral ($S = 0$), which means, on average, a duplication frequency in the population of 0.68–42% [= $\mu_c / (\mu_c + \mu_d)$]. By contrast, in presence of lactose, duplications are deleterious ($S \approx -28\%$), and then the average duplication frequency is of 0.09–0.11% [= $\mu_c / (\mu_c + \mu_d - S)$]. Note that the deletion rates are difficult to estimate experimentally, as this requires starting from a genotype with new-born (mostly unstable) duplications, albeit they are essential to properly understand the fixation process.

## Availability of resources

A Matlab code to model gene expression (*y*) and cell fitness (*W*) and a C++ code to perform the in silico evolution (as described above) are freely available for download at https://sourceforge.net/projects/rodrigo-duplications/files (*Rodrigo, 2017b*). A copy is archived at https://github.com/elifesciences-publications/rodrigo-duplications.

## Acknowledgements

This work was supported by grants BFU2015-66894-P (to GR) and BFU2015-66073-P (to MAF) from the Spanish Ministry of Economy (MINECO/FEDER), and also by grant GVA/2016/079 from the Generalitat Valenciana (to GR).

## Additional information

### Funding

| Funder | Grant reference number | Author |
|---|---|---|
| Ministerio de Economía y Competitividad | BFU2015-66894-P | Guillermo Rodrigo |
| Ministerio de Economía y Competitividad | BFU2015-66073-P | Mario A Fares |
| Generalitat Valenciana | GVA/2016/079 | Guillermo Rodrigo |

The funders had no role in study design, data collection and interpretation, or the decision to submit the work for publication.

### Author contributions

Guillermo Rodrigo, Conceptualization, Formal analysis, Validation, Methodology, Writing—original draft; Mario A Fares, Validation, Writing—original draft

### Author ORCIDs

Guillermo Rodrigo ⓘ https://orcid.org/0000-0002-1871-9617

### Decision letter and Author response

Decision letter https://doi.org/10.7554/eLife.29739.011
Author response https://doi.org/10.7554/eLife.29739.012

## Additional files

### Supplementary files

• Transparent reporting form
DOI: https://doi.org/10.7554/eLife.29739.009

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
