## [Decision Letter]

[Editors’ note: this article was originally rejected after discussions between the reviewers, but the authors were invited to resubmit after an appeal against the decision.]

Thank you for submitting your work entitled "Intrinsic adaptive value and early fate of gene duplication revealed by a bottom-up approach" for consideration by *eLife*. Your article has been reviewed by three peer reviewers, and the evaluation has been overseen by a Reviewing Editor and a Senior Editor. The following individuals involved in review of your submission have agreed to reveal their identity: Ashley Teufel (Reviewer #3).

Our decision has been reached after consultation between the reviewers. Based on these discussions and the individual reviews below, we regret to inform you that your work will not be considered further for publication in *eLife*.

As you will see, the referees found the problem to be interesting and appreciated your approach, but they also raised a number of issues which preclude publication in *eLife*. These include:

My read of the reviews is that while there might be an interesting glimmer of an idea here, it hasn't been cashed out properly and what we're left with is a not particularly persuasive model/argument.

It seems that the problems include:

1) An inadequate treatment of previous literature in the area;

2) A possible overstatement of the possible fitness benefits associated with reducing intrinsic noise;

3) Highly optimistic assumptions about how dosage compensation would work and how gene expression would be partitioned.

Reviewer #1:

Authors propose that gene duplication reduces gene expression noise, which could be beneficial and hence lead to the fixation of newly duplicated genes. However, I believe this model is untenable. My detailed comments follow.

1) There is no lack of evolutionary models to explain the fixation and long-term retention of duplicates. In terms of fixation, which is the focus of the present study, a new duplicate may be fixed by positive selection for increased gene dose (for either the main or minor function of the gene) or genetic drift. The Introduction seems to suggest a lack of suitable models (hence need for a new model), which is misleading.

2) The present model relies on a reduction of gene expression noise caused by gene duplication. This noise reduction is tiny. In the best case scenario, the intrinsic noise measured by CV is reduced by 29% upon duplication. But because expression noise is mainly from extrinsic noise, which is not reduced by gene duplication, the fraction of total expression noise reduced is minute (likely <5%).

3) The fitness benefit from 5% noise reduction will be swamped by a much greater fitness cost of doubling the expression of the gene owing to gene duplication. So, authors propose that the total expression level of the duplicate pair is halved by a mutation. Simply halving the total expression is actually not sufficient, because the above calculation of the noise reduction assumes equal expression levels of the two genes. If the two genes have different expression levels, the amount of noise reduced becomes even smaller. So, two very special mutations that reduce the expression of each gene by ~50% are required. I believe the probability of simultaneously acquiring two such mutations in a cell is almost 0.

4) Compared with the probability of acquiring the above two mutations, the probability of acquiring mutation(s) conferring a new function may be larger. In other words, neofunctionalization is probably more likely to happen than the scenario proposed by the authors.

5) Authors based their calculations on one gene (lacZ), but wrote as if the calculations apply to all genes.

6) I wonder why lacZ is not duplicated in *E. coli* if their theory predicts that duplication of lacZ is beneficial.

7) They provide no empirical evidence for even one duplicate gene that was likely fixed by the mechanism proposed.

Reviewer #2:

Rodrigo and Fares examine the immediate effect of gene duplication on gene expression and fitness, specifically on the reduction of noise in gene expression. The subject is of interest to a large community, including those interested in gene duplication, the evolution of regulatory systems and also microbial adaptation. The way to approach the problem is rather novel and brings to light new aspects on the issue of why would immediate gene duplication be maintained if dosage itself is not favored. The fact that gene duplication may reduce intrinsic noise in gene expression has been noticed before (Wang and Zhang, as cited by the authors) and directly derives from the fact the average of two random values from a distribution are closer to the mean of that distribution than any of the single values are, at least for the type of distributions we are dealing with. Showing this using a well-known regulatory system is valuable, especially if other factors such as trade-off that may derive from the cost of expression or the cost of having two gene copies are considered. That being said, the manuscript as presented would require additional work before it can be considered for publication. Here are some points:

1) The writing needs to be improved. Some wordings are confusing and also the overall structure of the paper would benefit from a more logical organization. The different alternative assumptions should be confronted directly side by side to clarify the limitations and the conditions in which the system could evolve. The issue of trade-off that is introduced as being important in the Introduction should be better addressed.

Here are some examples of sentences that need to be revisited:

- The first sentence of the Abstract is difficult to read. The fact that duplication contributes to complexity implies that duplicates are maintained. Why use “albeit”?

- Introduction, second paragraph, first sentence. This sentence is also hard to follow and brings multiple important elements in a single sentence.

- Introduction, fifth paragraph, second to last sentence. Again hard to follow.

- Introduction, last paragraph. What is a real gene?

- “Here, we simply considered a cost function based on LacZ expression, although it would be more precise a cost based on lactose permease (LacY) activity (Eames and Kortemme, 2012)”. This sentence is difficult to follow.

- “The population was let to evolve without introducing any artifact…”? What does artifact refer to here?

- Subsection “Most of the new-born duplicated genes are costly for the cell and do not offer phenotypic accuracy”, last paragraph. It is strange to start this paragraph with “even though”. We would expect a contrast to be made but it is not the case.

- Subsection “Most of the new-born duplicated genes are costly for the cell and do not offer phenotypic accuracy” “as long as” is not used in the proper context.

- Subsection “Fixation is conditioned by the unexpected recurrence of creation and deletion of gene duplications in a population”, first sentence. The word “created” is used to refer to gene duplication. Gene duplication is a process that cannot be created. Gene duplicates can be created, although I would refrain from using “created” here.

- The authors use “simple” and “punctual” mutation rates. They may want to refer to per base or nucleotide mutation rate instead, or use other standard nomenclature.

- Subsection “A comprehensive model compatible with population genetics to explain the early fate of gene duplications”, second paragraph. It would be better to state directly the effect of generation time rather than mention “simple” organisms because they have short generation time.

.…

2) A large fraction of the results presented assume that the sum of expression of the two copies is equal to the expression of the ancestral copy. This assumption is later relaxed in the paper. However, because expression is likely to scale with copy number, this assumption is most likely extremely optimistic. In addition, it is possible that with two copies, repression is not as efficient and the genes are now expressed even when not needed. The two different scenario (2X expression and 1X expression, and their intermediates if possible) should be compared side-by-side and better arguments should be presented as to why 1X is achievable.

3) The issue of intrinsic and extrinsic noises should be brought earlier in the paper as this is a very important consideration. They could be introduced in the Introduction. Gene duplication is not expected to reduce extrinsic noise and extrinsic noise is usually the primary source of differences among cells. As far as I understand, they are treated as potentially contributing equally in the model, which is clearly not the case in reality.

4) An alternative to duplication is also an increase in expression level, which would make protein abundance more often above the critical expression value and thus also increase fitness, without the need for duplication. Mutations that increase abundance would also then compete with duplications.

5) Subsection “Gene duplication helps to better resolve the fitness trade-off”, second paragraph. The authors describe the fitness landscape as rugged but a rugged fitness landscape has multiple local peaks, which is not the case here.

6) The authors define and introduce phenotypic accuracy, which is basically the inverse of noise. I am not sure more terms are necessary in this field. Not sure also that the use of information transmission helps this study and adds anything to the results.

7) Subsection “Gene duplication helps to better resolve the fitness trade-off”, last paragraph. The authors say that the two surfaces reassemble. This interpretation appears to be rather subjective. It would be useful to explain why this matters and how similar they really are.

8) The authors introduce the concept of trade-off in the Introduction and argue that this is an important factor in evolution but largely ignore them as a constraint on the evolution of expression. At the same time, they state that an increase in expression is detrimental in most environments (subsection “Most of the new-born duplicated genes are costly for the cell and do not offer phenotypic accuracy”, first paragraph). This question needs clarification and again, a better organization of the text would allow to better contrast the systems with and without trade-offs.

9) The authors use a biological context that is laboratory populations and experimental evolution. For instance, they say in the first paragraph of the subsection “Fixation is conditioned by the unexpected recurrence of creation and deletion of gene duplications in a population”, that typical bacteria populations are 10^9^ cells. I presume that they refer to cell populations in the laboratory. It would be more appropriate to refer to biological conditions that occur in nature. Even if laboratory conditions favor some evolutionary paths or dynamics, it would be irrelevant if the conditions do not exist in nature. This comment is also relevant for the simulations with dilutions and exponential growth in a flask. These simulations would be interesting if they were tested experimentally in the laboratory in this study. However, since we want to understand evolution in nature, why not use what is expected to be relevant in natural populations, including effective population size estimates, which have been computed for *E. coli* I presume. Since theory has shown that duplication reduces noise, what the readers will be really interested in is whether this is sufficient to favor the maintenance of duplicates in a biologically realistic system.

10) Subsection “Phenotypic accuracy can lead to the fixation of a new-born duplicated gene in the population”, first paragraph. Cis regulatory mutations are assumed to act on the average expression and not on the noise in expression. This is a convenient assumption but not necessarily true. Mutations most likely affect both at the same time (See the work of P. Wittkopp). This could reduce the mutational target site available for mutations reducing expression level.

11) Subsection “Phenotypic accuracy can lead to the fixation of a new-born duplicated gene in the population”, first paragraph. The authors discuss the fact that about 10% of mutations affect expression (reduction by about 50%). To calculate the rate at which these mutations occur, one needs to know how many sites in the genome have these effects, not what fraction of mutations that have been studied reduce expression. It is not 10% of all mutations in the genome that reduce expression, but rather 50% of the 75 bp region as studied by Kinney et al. This should be clarified in the calculation. Also, the probability that a duplication and a mutation that reduces expression by 50% occur in the same cell in the same generation depend on their equilibrium frequency and somehow the effective population size? The order of appearance would also matter because reducing the expression of only one of the copy (if the mutation occurs after the duplication) is not going to bring the expression level to the ancestral state.

12) Discussion, second paragraph. It is not clear that all of the results mentioned here derive from the theory proposed and if the results actually suggest a reinterpretation of the results. To be useful, it would be important to have predictions from this model that would be specific to this model and could not be explained by the previous models proposed. Also, it would be useful if some were tested here to actually show that some cases known in nature seem to fit the model. Any variation in gene copy number in bacteria that cannot be explained by dosage effects alone or other models of duplicate evolution?

13) The authors assume that the gene expression partitioning seen for pairs of duplicates is 50-50%, but according to Gout and Lynch, this is very often not the case. It is not clear how an expression partitioning that is not 50% contributes to reduce noise in expression. This could be explored here.

14) Some reasoning needs to be revisited carefully. For instance, in the third paragraph of the Discussion, the authors predict that essential genes would be less duplicable as a consequence of their reduced expression noise. Essential genes are not created essential and may derive from non-essential genes, which were noisy initially. If these genes show less expression noise because they contribute more to fitness, it means that selection for lower noise could have favoured their duplication (at the same time making them non-essential, making this effect hard to see).

15) Subsection “In silico evolution”. Why is evolution envisioned as if it occurred in the laboratory? It is already unclear if experimental evolution reflects evolution in natural systems so simulating experimental evolution appears to move away from nature.

16)Subsection “Genetic diversity”, last paragraph. Wouldn't the equilibrium frequency just be Uc/Ud?

17) Figure 1. Should explain what is x/x_0_ in the legend.

Reviewer #3:

This manuscript puts forth an interesting new theory on how newly birthed duplicated genes could eventually fix in a population. While the work laid out here seems to be of large interest, I have a few concerns that I would like to see addressed before publication.

My main concern with this publication is that the bulk of the work is centered on examining a system where a duplication does not result in a change of total expression, which is at best a very rare occurrence. While this is discussed later in the manuscript, some justification of why this situation was chosen for the biases of this work should be included in the "Gene duplication helps to better resolve the fitness trade-off" section.

The claim that the actual and the optimal dose-response curves (Figure 1 and Figure 2) are similar doesn't seem very convincing. Showing this data in something like a q-q plot and reporting a correlation would aid in the argument. This is especially important for Figure 2 when you make the comparison between the duplicate and the singleton, because there does not appear to be much of difference between both.

The comparison of non-normalized mutual information is confusing. Stating what the I values are and that one is 25% higher than the other doesn't convey the message that the duplication changes fidelity in a significant way. Is there an additional metric that could be used to better make this point?

The set up in the Introduction could be improved by adding further detail about why reducing gene expression inaccuracies results in increased fitness.

Often the model is linked to values that have been "experimentally determined" but there doesn't appear to be a clear reference to where these values have come from.

The amount of in. noise is an important parameter in this model. Any statement about the amount of in. noise that exists in biological systems would aid in linking this model back to the biology. Is a moderate (0.3) amount of in. noise to be expected?

The Discussion section largely centers on further directions of this work and ends abruptly. Including a section about the limitations of this work and also casting this work into a larger context would be appreciated.

I believe that *eLife* requires that you make any code used available. I would suggest putting your simulation code in a repository and including the link in the manuscript.

Overall, this is an interesting manuscript but I feel the way some of the data is presented could be changed to strengthen the author's arguments. By including more detailed statistical analysis and expanding some portions of text for clarity would improve this manuscript substantially.

[Editors’ note: what now follows is the decision letter after the authors submitted for further consideration.]

Thank you for resubmitting your work entitled "Intrinsic adaptive value and early fate of gene duplication revealed by a bottom-up approach" for further consideration at *eLife*. Your revised article has been favorably evaluated by Diethard Tautz as the Senior and Reviewing Editor, and two reviewers.

The manuscript has been improved but there are some remaining issues that need to be addressed before acceptance, as outlined below:

One reviewer still has some comments on the presentation of your results. Please check these carefully and clarify the issues as much as possible. Such changes should improve the impact that this manuscript will eventually have. While I do not think that further reviewing will be necessary after these changes are introduced, I should like to ask you to provide nonetheless a careful response letter, indicating which changes were incorporated.

Reviewer #2:

I maintain my comments on the previous version of the manuscript. I believe the paper is hard to follow and extremely specialized such that it is hard to evaluate whether the observations are generalizable.

Important concepts in some sections are not introduced properly in the Introduction (tradeoffs for instance do not only include production costs but any other types of negative effects, including in other environments). Some assumptions made for the analysis are not well detailed, for instance the extent of noise in expression, the cost of expression. Another example is the statement made from Figure 6 that most mutations are nearly neutral. Given what was said about the importance of gene expression tuning and the large Ne for *E. coli*, most of these changes are most likely not neutral at all. This is a surprising statement given that the paper shows that small changes in the distribution of expression levels can affect the fate of gene duplication.

What we would like to know is under which noise regime (showed to be likely based on observations) this mechanism could affect evolution given a clear set of assumptions that are shown to be realistic. I do not feel we know this by reading the paper as it is.

Some of the concepts introduced is not defined properly, for instance phenotypic accuracy. Here the authors say that phenotypic accuracy (…subsection “Gene duplication helps to better resolve the fitness trade-off”, second paragraph) is the fact that phenotypic responses generated by duplicated genes give on average higher fitness values than responses generated by singletons. This is simple corollary to the fact that duplication reduces noise, this is not a new concept that needs the be defined. Using such definitions is just a distraction that reduces our understanding. Same could be said about information content. This is not appropriate for a generalist journal such as *eLife*.

It is not clear why we need simulations at all if the selection coefficient have been estimated given all of the analytical work that has been done previously (fixation prob. versus Ne and S).

The section on expression demand in extreme environments (subsection “Expression demand in extreme environments can also lead to the fixation of a newborn duplicate in the population”) does not really deal with the question in hand, which is the effect of duplication of noise reduction. There are examples of arbitrary assumptions here too, for instance the consideration of a lac promoter with 40% lower activity as a starting point.

Examples of sections with lack of logical flow:

Introduction, fifth paragraph; subsection “Gene duplication helps to better resolve the fitness trade-off”, first two paragraphs; subsection “Expression demand in extreme environments can also lead to the fixation of a newborn duplicate in the population”, second paragraph.

Reviewer #3:

This version of the manuscript is much improved. I thank the author for careful and detailed comments. I especially appreciate the inclusion of significance statistics and the addition of the "Maintenance of a duplication upon fixation in the population" section.

---

## [Author Response]

[Editors’ note: the author responses to the first round of peer review follow.]

Reviewer #1:Authors propose that gene duplication reduces gene expression noise, which could be beneficial and hence lead to the fixation of newly duplicated genes. However, I believe this model is untenable. My detailed comments follow.1) There is no lack of evolutionary models to explain the fixation and long-term retention of duplicates. In terms of fixation, which is the focus of the present study, a new duplicate may be fixed by positive selection for increased gene dose (for either the main or minor function of the gene) or genetic drift. The Introduction seems to suggest a lack of suitable models (hence need for a new model), which is misleading.

We do not agree with this appreciation, as saying that fixation of gene duplicates is well explained just by selection for increased dosage or by drift is ignoring a lot of literature. The search for alternative models date back several decades. As already pointed out by Spofford (Am. Nat. 103:407-432, 1969), there is a significant gap in our understanding of gene duplication concerning the critical initial phase; a gap that still needs to be filled. Please, see the review by Innan and Kondrashov, 2010 where several models are discussed regarding this long-standing problem. Our work indeed goes in this direction, trying to determine if duplication is advantageous per se, without invoking the necessity of more expression or a new function. At this point, we would like to stress that our focus is on the earliest stages of selection on gene duplication.

Many records expose that dosage or drift are not sufficient criteria. For example, an increased expression can be useful in extreme circumstances (e.g., to face a stress), but not in routine environments for which the organism should be adapted. Moreover, many duplicates do not contribute to increase total expression (Qian et al., 2010), which entails they were not selected for dosage. It is certainly unlikely that all these duplicates were fixed fortuitously, so additional criteria are required. We believe that all this is well explained in the Introduction, so we ask reviewer to reconsider his/her position after re-reading the manuscript.

But even more. Selection for increased dosage or by drift is a model that cannot explain, for example, why duplicates maintain expression, as well as function (DeLuna et al., 2008), or why duplicates are enriched in TATA boxes, elements that contribute to increase variability (Lehner, 2010). These relevant aspects are discussed in our manuscript, and our results indeed contribute to enlarge our knowledge about gene expression control mechanisms, and in particular the mechanism of gene duplication.

2) The present model relies on a reduction of gene expression noise caused by gene duplication. This noise reduction is tiny. In the best case scenario, the intrinsic noise measured by CV is reduced by 29% upon duplication. But because expression noise is mainly from extrinsic noise, which is not reduced by gene duplication, the fraction of total expression noise reduced is minute (likely <5%).

We are sorry to say these numbers are wrong.The variance of the stochastic fluctuations in gene expression is reduced by 50% upon duplication when only intrinsic noise is taken into account. This leads to an average selection coefficient of 0.1% (duplicate vs. singleton), which is sufficient to be selected (Figure 4). When in addition extrinsic noise is considered, the stochastic fluctuations in gene expression are reduced in a lesser, but still significant way upon duplication, leading to selection coefficients of about 0.05%, which is still selectable in populations larger than 10^4^ individuals (Figure 4). When intrinsic and extrinsic noise are of similar magnitude, the reduction is of 25%. When extrinsic noise is double the intrinsic noise, the reduction is of 17%. We have rewritten the text to have: “the variance of the stochastic fluctuations (noise) in gene expression is reduced by 50% upon duplication (Wang and Zhang, 2011; when only intrinsic fluctuations are considered). However, when both intrinsic and extrinsic fluctuations are considered, the variance is reduced by 15 – 25%. In any case, this increases fitness on average”. After this clarification, we hope reviewer now agrees with us.

3) The fitness benefit from 5% noise reduction will be swamped by a much greater fitness cost of doubling the expression of the gene owing to gene duplication. So, authors propose that the total expression level of the duplicate pair is halved by a mutation. Simply halving the total expression is actually not sufficient, because the above calculation of the noise reduction assumes equal expression levels of the two genes. If the two genes have different expression levels, the amount of noise reduced becomes even smaller. So, two very special mutations that reduce the expression of each gene by ~50% are required. I believe the probability of simultaneously acquiring two such mutations in a cell is almost 0.

As said before, the reviewer’s claim of 5% noise reduction is erroneous. That being said, the model only requires one mutation, so the second reviewer’s claim here about two mutations is also erroneous. Cell populations inherently present genetic variability, and in few generations one could find a mutant with half expression. The idea is that duplication occurs in that mutant cell. The precise probabilities of occurrence and fixation are calculated in this work. Note that we have recalculated them in the new version following the suggestion of reviewer 2.

4) Compared with the probability of acquiring the above two mutations, the probability of acquiring mutation(s) conferring a new function may be larger. In other words, neofunctionalization is probably more likely to happen than the scenario proposed by the authors.

As stated before, the model is not based on the accumulation of two mutations, but just one. Therefore, this comment does not apply.

5) Authors based their calculations on one gene (lacZ), but wrote as if the calculations apply to all genes.

Indeed. The theory is general and is not particular of any gene. We chose *lacZ* gene for illustrative purposes and because the dose-response curve and associated fitness function are known in this case (experimentally validated). The *lac* operon has been studied since the times of Jacob and Monod as a paradigmatic system from which to derive general principles of gene regulation. See e.g. the works by Elowitz et al., 2002 to study stochastic gene expression or by Garcia and Phillips (PNAS 108:12173-12178, 2011) to study thermodynamics of transcriptional repression, both with the *lac* promoter. This is normal practice in the field of dynamic systems biology, where the detailed study of the dynamic behavior of natural systems is able to derive general rules applicable to any system with similar molecular features (see the Alon’s textbook). In words of Savageau, “the lactose (*lac*) operon of *Escherichia coli* serves as the paradigm for gene regulation, not only for bacteria, but also for all biological systems from simple phage to humans. The details of the systems may differ, but the key conceptual framework remains, and the original system continues to reveal deeper insights with continued experimental and theoretical study” (Math. Biosci. 231:1938, 2011).

6) I wonder why lacZ is not duplicated in E. coli if their theory predicts that duplication of lacZ is beneficial.

As we discuss in the manuscript, duplicates are created and deleted continuously (subsection “Fixation is conditioned by the unexpected recurrence of formation and deletion of duplicates in a population”). Because these rates have been shown to be higher than expected (see e.g. Genetics 184:1077-1094, 2010 or Genetics 194:937-954, 2013), the selection coefficient has to be greater than the deletion rate to escape from such birth-death processes and reach fixation. In bacteria, the deletion rate for *lacZ* gene is quite high (~10^-2^, Genetics 184:1077-1094, 2010), which is greater than the average selection coefficient that we calculated (0.1% = 10^-3^), then preventing its duplication. For other genes (or the same *lacZ* gene in other genetic context), the deletion rate can be much lower (~10^-4^, Genetics 192:397-415, 2012), which is now smaller than the selection coefficient and fixation of a duplicate can occur. This relevant aspect has been missed by this reviewer. To reinforce this, we have added the following sentence: “perhaps, this is why lacZ is not duplicated in *E. coli* despite this may be beneficial”.

Certainly, we are not interested in analyzing the duplicability of particular genes, but in providing a general theory to explain the intrinsic adaptive value of duplicates, something that completes the model of fixation by dose selection or drift (Figure 7).

7) They provide no empirical evidence for even one duplicate gene that was likely fixed by the mechanism proposed.

We have added the following text in the new version of the manuscript: “The genomic inspection of organisms in which genetic drift is not, in principle, a suitable force to drive the fixation of duplicates (e.g., bacteria or yeast; Lynch and Marinov, 2015) gave us some empirical insight, despite the masking produced by subsequent evolutionary trajectories. […] Arguably, duplication might have been fixed in this case to cope with gene expression inaccuracies, especially when GntR produces bimodal responses (captured in single-cell experiments; Afroz et al., 2014)”.

This contains some analysis of duplicated genes in bacteria and yeast to support our theory.

Reviewer #2:[…] 1) The writing needs to be improved. Some wordings are confusing and also the overall structure of the paper would benefit from a more logical organization. The different alternative assumptions should be confronted directly side by side to clarify the limitations and the conditions in which the system could evolve. The issue of trade-off that is introduced as being important in the Introduction should be better addressed.Here are some examples of sentences that need to be revisited:- The first sentence of the Abstract is difficult to read. The fact that duplication contributes to complexity implies that duplicates are maintained. Why use “albeit”?

Removed.

- Introduction, second paragraph, first sentence. This sentence is also hard to follow and brings multiple important elements in a single sentence.

Split.

- Introduction, fifth paragraph, second to last sentence. Again hard to follow.

Rewritten.

- Introduction, last paragraph. What is a real gene?

“Real” removed.

- “Here, we simply considered a cost function based on LacZ expression, although it would be more precise a cost based on lactose permease (LacY) activity (Eames and Kortemme, 2012)”. This sentence is difficult to follow.

Rewritten.

- “The population was let to evolve without introducing any artifact…”? What does artifact refer to here?

Replaced by “bias”.

- Subsection “Most of the new-born duplicated genes are costly for the cell and do not offer phenotypic accuracy”, last paragraph. It is strange to start this paragraph with “even though”. We would expect a contrast to be made but it is not the case.

Removed.

Subsection “Most of the new-born duplicated genes are costly for the cell and do not offer phenotypic accuracy” “as long as” is not used in the proper context.

Restructured.

- Subsection “Fixation is conditioned by the unexpected recurrence of creation and deletion of gene duplications in a population”, first sentence. The word “created” is used to refer to gene duplication. Gene duplication is a process that cannot be created. Gene duplicates can be created, although I would refrain from using “created” here.

Fixed.

- The authors use “simple” and “punctual” mutation rates. They may want to refer to per base or nucleotide mutation rate instead, or use other standard nomenclature.

Fixed.

- Subsection “A comprehensive model compatible with population genetics to explain the early fate of gene duplications”, second paragraph. It would be better to state directly the effect of generation time rather than mention “simple” organisms because they have short generation time.

Fixed.

2) A large fraction of the results presented assume that the sum of expression of the two copies is equal to the expression of the ancestral copy. This assumption is later relaxed in the paper. However, because expression is likely to scale with copy number, this assumption is most likely extremely optimistic. In addition, it is possible that with two copies, repression is not as efficient and the genes are now expressed even when not needed. The two different scenario (2X expression and 1X expression, and their intermediates if possible) should be compared side-by-side and better arguments should be presented as to why 1X is achievable.

Our model indeed considers that expression is doubled after duplication. Thus, it does scale with copy number, and we never assumed the contrary. The relevant question is, hence, how this scenario is compatible with dosage sharing. We solved this by proposing that the genetic variability existing in a population allows having mutants with half expression (see Figure 6), which can recover the ancestral expression level after duplication. The effect of intermediates between 1x and 2x is shown in Figure 6.

In the first part of the manuscript, we assumed that a genotype carrying a duplicate with dosage sharing already existed to study its eventual selective advantage. We have added the following sentence: “For the moment, we ensured gene dosage sharing to evaluate in a quantitative way the goodness of having a second gene copy for the cell without invoking the need for more expression”. In the second part, we studied how to reach such genotype, calculating the precise probabilities of occurrence and fixation. We have added new figures to elucidate our point (Figure 2, Figure 5, Figure 6 in the new version).

In the particular case of the *lac* promoter, repression is highly strong (the effective dissociation constant between LacI and DNA is about nM, i.e., a few LacI molecules are sufficient to turn off completely the gene). Although repression can change after duplication when it is weak, this effect can be neglected to develop a first theory.

3) The issue of intrinsic and extrinsic noises should be brought earlier in the paper as this is a very important consideration. They could be introduced in the Introduction. Gene duplication is not expected to reduce extrinsic noise and extrinsic noise is usually the primary source of differences among cells. As far as I understand, they are treated as potentially contributing equally in the model, which is clearly not the case in reality.

Although the Introduction is already quite long, we have expanded it to include more detail about gene expression noise. The balance between intrinsic and extrinsic noise depends on the particular environmental conditions and the regulatory structures in which the gene is embedded.

In some cases, extrinsic noise is dominant, in others not (see e.g. PNAS 99:12795-12800, 2002 and Nature 439:861-864, 2006). For example, when the medium is rich in nutrients, the expression is low, and no further regulations affect the gene, intrinsic noise dominates. However, when the medium is poor, the expression is high, and the gene belongs to a complex regulatory network, extrinsic noise does. According to our results, when only intrinsic noise is considered, genetic variability is reduced in about 50%, while when both intrinsic and extrinsic noises are considered it is reduced in about 25%. But extrinsic noise does not seem to affect much the selective advantage provided by duplication (see Figure 4).

Moreover, if reviewer looks carefully at the model, he/she will notice that intrinsic and extrinsic noises are treated differently. Intrinsic noise is particular for each copy (z1 and z2), while extrinsic noise is common for both copies (z0). Thus, our model indeed reflects what happens in reality.

4) An alternative to duplication is also an increase in expression level, which would make protein abundance more often above the critical expression value and thus also increase fitness, without the need for duplication. Mutations that increase abundance would also then compete with duplications.

This is precisely the study that we did, results shown in Figure 6. See the associated text in the subsection “Expression demand in extreme environments can also lead to the fixation of a newborn duplicate in the population”.

5) Subsection “Gene duplication helps to better resolve the fitness trade-off”, second paragraph. The authors describe the fitness landscape as rugged but a rugged fitness landscape has multiple local peaks, which is not the case here.

We have replaced “rugged” by “hill-like” in the new version.

6) The authors define and introduce phenotypic accuracy, which is basically the inverse of noise. I am not sure more terms are necessary in this field. Not sure also that the use of information transmission helps this study and adds anything to the results.

Although we consider relevant the introduction of “phenotypic accuracy”, as it is almost a self-explanatory term, we have diminished its use in the manuscript. This is putting the system in an operational point close to the optimum, that is, make the phenotype as accurate as possible. This is more than noise reduction (subsection “Gene duplication helps to better resolve the fitness trade-off”).

On the other hand, the use information transmission further characterizes noise reduction for varying lactose dose. It is true it could be overlooked, although it brings a single parameter able to describe the accuracy of the dynamic response. In addition, this follows the tradition of using this information theoretic parameter to characterize regulatory systems with acquired genetic redundancy (Science 334:354-358, 2011, *eLife* 4:e06559, 2015, or PLoS Comput. Biol. 12:e1005156, 2016).

7) Subsection “Gene duplication helps to better resolve the fitness trade-off”, last paragraph. The authors say that the two surfaces reassemble. This interpretation appears to be rather subjective. It would be useful to explain why this matters and how similar they really are.

The two surfaces resemble in the sense that they present the maximum for the same lactose dose. We performed a Spearman’s correlation test. This matters because it shows that mutual information (simply measured from gene expression data) can be used as a proxy of selection coefficient (hard to measure). We have added the following sentence: “We could then predict a cell fitness increment from measuring a reduction in gene expression noise”.

8) The authors introduce the concept of trade-off in the Introduction and argue that this is an important factor in evolution but largely ignore them as a constraint on the evolution of expression. At the same time, they state that an increase in expression is detrimental in most environments (subsection “Most of the new-born duplicated genes are costly for the cell and do not offer phenotypic accuracy”, first paragraph). This question needs clarification and again, a better organization of the text would allow to better contrast the systems with and without trade-offs.

We do not understand these statements. The trade-off, as stated in the manuscript (subsection “Quantitative biochemical view of a fitness trade-off”), arises because the enzyme (LacZ) generates at the same time a benefit (in metabolism) and a cost (due to expression). See Figure 1. The model always takes into account the balance between metabolic benefit and expression cost (i.e., the trade-off). There are no simulations in which the expression cost in neglected. Stochastic fluctuations in expression are evaluated on the basis of such trade-off, as well as changes in mean expression.

The central aspect of this work is now illustrated in Figure 2. When the system is close to the optimal operational point (i.e., maximal fitness), changes in gene expression are costly (i.e., reduce fitness). These changes can be stochastic due to molecular noise or deterministic. Thus, noise reduction by gene duplication results in a useful strategy to increase fitness, provided the total gene expression is maintained (Figure 3). If gene expression were increased after duplication, the system would move away from the optimum, with the consequent reduction in fitness (Figure 5). This is why we proposed a model in which a mutation previous duplication is required. However, when the system is far from the optimum, changes in gene expression can be favorable (Figure 6). This is the case in which duplication is selected by dosage.

9) The authors use a biological context that is laboratory populations and experimental evolution. For instance, they say in the first paragraph of the subsection “Fixation is conditioned by the unexpected recurrence of creation and deletion of gene duplications in a population”, that typical bacteria populations are 10^9^ cells. I presume that they refer to cell populations in the laboratory. It would be more appropriate to refer to biological conditions that occur in nature. Even if laboratory conditions favor some evolutionary paths or dynamics, it would be irrelevant if the conditions do not exist in nature. This comment is also relevant for the simulations with dilutions and exponential growth in a flask. These simulations would be interesting if they were tested experimentally in the laboratory in this study. However, since we want to understand evolution in nature, why not use what is expected to be relevant in natural populations, including effective population size estimates, which have been computed for E. coli I presume. Since theory has shown that duplication reduces noise, what the readers will be really interested in is whether this is sufficient to favor the maintenance of duplicates in a biologically realistic system.

In this new version, we have recalculated the probability of fixation with the value of the effective population size for *E. coli* (about 10^8^).

10) Subsection “Phenotypic accuracy can lead to the fixation of a new-born duplicated gene in the population”, first paragraph. Cis regulatory mutations are assumed to act on the average expression and not on the noise in expression. This is a convenient assumption but not necessarily true. Mutations most likely affect both at the same time (See the work of P. Wittkopp). This could reduce the mutational target site available for mutations reducing expression level.

This is a very interesting remark. We have introduced a citation to Metzger et al. (Nature 521:344-347, 2015) in this new version. We now say: “Mutations were also assumed to affect only the mean expression level and not the noise, even though this latter might happen (Metzger et al., 2015)”. Also in the Discussion, “These predictions involve, nevertheless, some limitations. On the one hand, due to a simplified mathematical model not considering the many molecular/genetic attributes that impinge implicitly on gene expression, such as promoter sequence-dependent noise levels (Metzger et al., 2015), response coupling due to genetic proximity (Becskei et al., 2005), or recursive fitness-expression dependence (Klumpp et al., 2009)”.

11) Subsection “Phenotypic accuracy can lead to the fixation of a new-born duplicated gene in the population”, first paragraph. The authors discuss the fact that about 10% of mutations affect expression (reduction by about 50%). To calculate the rate at which these mutations occur, one needs to know how many sites in the genome have these effects, not what fraction of mutations that have been studied reduce expression. It is not 10% of all mutations in the genome that reduce expression, but rather 50% of the 75 bp region as studied by Kinney et al. This should be clarified in the calculation. Also, the probability that a duplication and a mutation that reduces expression by 50% occur in the same cell in the same generation depend on their equilibrium frequency and somehow the effective population size? The order of appearance would also matter because reducing the expression of only one of the copy (if the mutation occurs after the duplication) is not going to bring the expression level to the ancestral state.

There, we were talking about the mutations falling in the promoter (as the sentence starts with “Mutations in the cis-regulatory region of the lacZ gene…”). The 10% refers to a 75 bp region, not the whole genome.

We would say that the probability of co-occurrence (a mutation that reduces in half expression and a duplication) is independent of the population size. The population size determines the number of generations to wait for such co-occurrence.

The order of appearance of these mutational events is certainly an aspect that we did not consider. This would reduce in half the estimation. We have amended this.

12) Discussion, second paragraph. It is not clear that all of the results mentioned here derive from the theory proposed and if the results actually suggest a reinterpretation of the results. To be useful, it would be important to have predictions from this model that would be specific to this model and could not be explained by the previous models proposed. Also, it would be useful if some were tested here to actually show that some cases known in nature seem to fit the model. Any variation in gene copy number in bacteria that cannot be explained by dosage effects alone or other models of duplicate evolution?

We have rewritten the Discussion to better provide an interpretation of our results. In addition, we have added text in the Results (subsection “Maintenance of a duplicate upon fixation in the population”), regarding the “genomic inspection of organisms”, to support a fixation model in which noise reduction is relevant. We have added accordingly significant references.

13) The authors assume that the gene expression partitioning seen for pairs of duplicates is 50-50%, but according to Gout and Lynch, this is very often not the case. It is not clear how an expression partitioning that is not 50% contributes to reduced noise in expression. This could be explored here.

This is precisely what we did in former Figure 6, citing in the text Gout and Lynch, 2015.

14) Some reasoning needs to be revisited carefully. For instance, in the third paragraph of the Discussion, the authors predict that essential genes would be less duplicable as a consequence of their reduced expression noise. Essential genes are not created essential and may derive from non-essential genes, which were noisy initially. If these genes show less expression noise because they contribute more to fitness, it means that selection for lower noise could have favoured their duplication (at the same time making them non-essential, making this effect hard to see).

We have added this interesting remark in the Discussion: “However, this consideration should be taken with caution, as genes not essential a priori could be duplicated and then, upon fixation, accumulate beneficial mutations (Han et al., 2009) to ensure preservation for long time, resulting a posteriori in essential genes due to functional diversification (as it seems in the case of mammals; Makino et al., 2009)”.

15) Subsection “In silico evolution”. Why is evolution envisioned as if it occurred in the laboratory? It is already unclear if experimental evolution reflects evolution in natural systems so simulating experimental evolution appears to move away from nature.

Experimental evolution was simulated as a way to determine whether a duplicate, increasing fitness on average due to noise reduction, would be able to invade a population. Invasion is not obvious because the fitness increase upon duplication occurs only on average. Of course, this framework is a proxy of what occurs in nature. But we consider it is sufficient for our purposes. We have acknowledged this aspect in the manuscript: “For simplicity, we simulated a scenario of experimental evolution (Elena and Lenski, 2003; Dekel and Alon, 2005), although the dynamics in nature might be more complex”.

16)Subsection “Genetic diversity”, last paragraph. Wouldn't the equilibrium frequency just be Uc/Ud?

No, the frequency is given by Uc/(Uc+Ud). Uc/Ud gives the ratio. If Ud>>Uc, then the frequency and the ratio can be considered equal.

17) Figure 1. Should explain what is x/x_0_ in the legend.

This corresponds to a normalized lactose concentration. This has been specified in the legend.

Reviewer #3:This manuscript puts forth an interesting new theory on how newly birthed duplicated genes could eventually fix in a population. While the work laid out here seems to be of large interest, I have a few concerns that I would like to see addressed before publication.My main concern with this publication is that the bulk of the work is centered on examining a system where a duplication does not result in a change of total expression, which is at best a very rare occurrence. While this is discussed later in the manuscript, some justification of why this situation was chosen for the biases of this work should be included in the "Gene duplication helps to better resolve the fitness trade-off" section.

The seminal study by Qian et al. (Trends Genet. 26:425-430, 2010) showed just the contrary. In fact, many old duplicates that were fixed still maintain a high degree of functional similarity, and each copy shows a substantial decrease in its expression level after duplication to maintain a similar level with respect to the ancestral genotype (dosage sharing model). In addition, recent works also appear to support this hypothesis, e.g., the ones by Lynch (Mol. Biol. Evol. 32:2141-2148, 2015) and Pritchard (Science 352:1009-1013, 2016).

That being said, our model indeed considers that expression is doubled after duplication. The relevant question is, hence, how this scenario is compatible with dosage sharing. We solved this by proposing that the genetic variability existing in a population allows having mutants with half expression (see Figure 6), which can recover the ancestral expression level after duplication. This is one of the key aspects of this work, and we hope now the reviewer realizes about it.

The claim that the actual and the optimal dose-response curves (Figure 1 and Figure 2) are similar doesn't seem very convincing. Showing this data in something like a q-q plot and reporting a correlation would aid in the argument. This is especially important for Figure 2 when you make the comparison between the duplicate and the singleton, because there does not appear to be much of difference between both.

A q-q plot does not apply here, as we are not dealing with probability distributions. In any case, we could use the Euclidean distance to measure the distance between the two curves. We have added in the new version the following: “By generating different dose-response curves with values of x_0_ (lactose EC_50_ on LacZ) between 0.01 and 1 mM, we found that most of them deviate from the optimal one (P = 0.02; Euclidean distance as a metric)”.

But the problem with this comment is that the reviewer seems to ignore the nature of the two curves put into question. The actual dose-response curve between lactose and LacZ is well described by a sigmoidal function in which the EC_50_ value is represented by the parameter x_0_ in our model. This curve was obtained just by fitting the resulting LacZ values against the different lactose doses. The optimal dose-response curve, by contrast, was obtained mathematically by derivation of the fitness function (i.e., dW/dy = 0); a fitness function constructed by Dekel and Alon, 2005 from the experimental evaluation of the metabolic benefit and the expression cost, not involving the parameter x_0_. In this regard, that both curves roughly overlap is remarkable, suggesting that the *lac* promoter evolved to reach optimality (indeed, the main result by Dekel and Alon). We relied on this model to develop of theoretical work, but we cannot re-explain all that study here.

The comparison of non-normalized mutual information is confusing. Stating what the I values are and that one is 25% higher than the other doesn't convey the message that the duplication changes fidelity in a significant way. Is there an additional metric that could be used to better make this point?

Information transfer is a well-known magnitude in information theory to describe input-output associations, introduced by Shannon several decades ago. In recent years, this magnitude is being used in biology to describe genetic systems [see e.g. the works by Levchenko (Science 334:354-358, 2011) or by O’Shea (*eLife* 4:e06559, 2015)], although it is still unfamiliar for many researchers. In this case, information transfer (measured as mutual information) captures all stochasticity underlying gene expression data to produce an outcome that be compared. An increase of 25% in mutual information is indeed significant. We have added in the new version the following: “significance assessed by a z-test, P = 0 with 10^4^ points” (Results) and “To compare statistically two I values, we followed the approximation proposed by Cellucci et al., (2005) to obtain an equivalent correlation coefficient, and then the Fisher’s r-to-z transformation” (Materials and methods).

Perhaps, the reviewer is more familiar with other measures of association, like Pearson’s correlation. Mutual information outperforms this kind of correlation (see e.g. PLoS Comput. Biol. 12:e1005156, 2016). In fact, the calculation of mutual information is equivalent to calculating the G statistic, a likelihood-ratio statistic (see the Sokal’s textbook).

The set up in the Introduction could be improved by adding further detail about why reducing gene expression inaccuracies results in increased fitness.

Although the Introduction is already quite long, we have expanded it to include more detail about gene expression noise.

Often the model is linked to values that have been "experimentally determined" but there doesn't appear to be a clear reference to where these values have come from.

As it is said in the Materials and methods section, the parameters that define the fitness function come from the work by Dekel and Alon (Nature 436:588-592, 2005).

The amount of in. noise is an important parameter in this model. Any statement about the amount of in. noise that exists in biological systems would aid in linking this model back to the biology. Is a moderate (0.3) amount of in. noise to be expected?

The values of noise that we considered in this study are based on previous studies analyzing by means of single-cell techniques gene expression variability. We have added the following: “Typical values characterizing the magnitude of the stochastic fluctuations (*η*_in_ and *η*_ex_) range between 0.1 and 0.5. They lead to values of gene expression noise (understood as the coefficient of variation) between 0.26 and 0.72 (in the case of *η*_in_ = *η*_ex_ and *x* = *x*_0_), in agreement with experimental reports (Elowitz et al., 2002)”.

The Discussion section largely centers on further directions of this work and ends abruptly. Including a section about the limitations of this work and also casting this work into a larger context would be appreciated.

We have expanded the Discussion. Note however that Figure 7 already puts our work in context.

I believe that eLife requires that you make any code used available. I would suggest putting your simulation code in a repository and including the link in the manuscript.

Regarding the fitness and gene expression models, we detailed the precise equations in the Materials and methods section, as well as all parameter values. In fact, the fitness function was developed by Dekel and Alon, 2005, as stated in the manuscript. Any reader could implement this model easily, as we did. Regarding the in silico evolution experiments, we implemented our own code to evolve a cell population according to a framework of experimental evolution (serial dilutions). With the instructions we provide in the manuscript, it is straightforward to reproduce our results. Anyway, in looking for acceptance in *eLife* of the manuscript, we have deposited in SourceForge a Matlab file to model the fitness and gene expression as a function of lactose, and a C++ file to perform the in silico evolution of a population of cells according to such fitness function.

Overall, this is an interesting manuscript but I feel the way some of the data is presented could be changed to strengthen the author's arguments. By including more detailed statistical analysis and expanding some portions of text for clarity would improve this manuscript substantially.

The reviewer must note that this is not a bioinformatic study in which several statistical analyses are carried out, but a dynamical systems study (a mathematical approach to evolutionary systems biology). Here, we relied on bottom-up models of stochastic gene expression and cell fitness to perform our calculations. Suggesting typical comments found in evaluations of bioinformatic studies (like doing more statistical analyses, correlation plots, etc.) seems not appropriate here.

[Editors’ note: the author responses to the re-review follow.]

The manuscript has been improved but there are some remaining issues that need to be addressed before acceptance, as outlined below:One reviewer still has some comments on the presentation of your results. Please check these carefully and clarify the issues as much as possible. Such changes should improve the impact that this manuscript will eventually have. While I do not think that further reviewing will be necessary after these changes are introduced, I should like to ask you to provide nonetheless a careful response letter, indicating which changes were incorporated.Reviewer #2:I maintain my comments on the previous version of the manuscript. I believe the paper is hard to follow and extremely specialized such that it is hard to evaluate whether the observations are generalizable.

We have followed all these suggestions to improve readability.

Important concepts in some sections are not introduced properly in the Introduction (tradeoffs for instance do not only include production costs but any other types of negative effects, including in other environments). Some assumptions made for the analysis are not well detailed, for instance the extent of noise in expression, the cost of expression. Another example is the statement made from Figure 6 that most mutations are nearly neutral. Given what was said about the importance of gene expression tuning and the large Ne for E. coli, most of these changes are most likely not neutral at all. This is a surprising statement given that the paper shows that small changes in the distribution of expression levels can affect the fate of gene duplication.

We have added the following in the first Results section: “In cellular systems, fitness trade-offs arise because beneficial actions involve costs. […] Such components can be described in different ways according to the problem. A paradigmatic and simple fitness trade-off emerges when…”.

Regarding the extent of noise, we have added: “The magnitudes of the stochastic fluctuations were chosen as to end in typical variations of lactose EC_50_ of 10 – 100%, up or down, resulting in values of gene expression noise, around 0.5, compatible with experimental results (Elowitz et al., 2002).” This information was specified in the Materials and methods section, but now it is also included in the Results section.

Regarding the cost of expression, we have added: “with a marginal cost of 0.036 in the units of the model (Dekel and Alon, 2005)”.

The sentence about mutations in the promoter has been rewritten as: “This indicates that about 10% of them yield cells with nearby 50% lower expression” (removing the claim about neutrality).

What we would like to know is under which noise regime (showed to be likely based on observations) this mechanism could affect evolution given a clear set of assumptions that are shown to be realistic. I do not feel we know this by reading the paper as it is.

We have introduced the following text in the Discussion section: “Certainly, by aggregating the responses of two genes, intrinsic fluctuations can be mitigated, but not fluctuations of extrinsic nature. […] To follow our model, noise has to mainly impinge the regulation of the system, i.e., disturb the link between the signal molecule and gene expression (Blake et al., 2006)”.

Some of the concepts introduced is not defined properly, for instance phenotypic accuracy. Here the authors say that phenotypic accuracy (subsection “Gene duplication helps to better resolve the fitness trade-off”, second paragraph) is the fact that phenotypic responses generated by duplicated genes give on average higher fitness values than responses generated by singletons. This is simple corollary to the fact that duplication reduces noise, this is not a new concept that needs the be defined. Using such definitions is just a distraction that reduces our understanding. Same could be said about information content. This is not appropriate for a generalist journal such as eLife.

We have avoided the introduction of the term “phenotypic accuracy”.

It is not clear why we need simulations at all if the selection coefficient have been estimated given all of the analytical work that has been done previously (fixation prob. versus Ne and S).

As noise reduction is a strategy that works on average (i.e., to observe its effects we need sampling), we believe that simulations (Figure 4) are interesting. It is not the same thing to have a constant selection coefficient (a cell with duplication performs better all the time) than an average selection coefficient (a cell with duplication performs better most of the time, but not all).

The section on expression demand in extreme environments (subsection “Expression demand in extreme environments can also lead to the fixation of a newborn duplicate in the population”) does not really deal with the question in hand, which is the effect of duplication of noise reduction. There are examples of arbitrary assumptions here too, for instance the consideration of a lac promoter with 40% lower activity as a starting point.

We have removed this section (text and corresponding figures) from the manuscript.

Examples of sections with lack of logical flow:Introduction, fifth paragraph;

Rewritten as: “But this strategy works on average, i.e., duplication may warrant more accuracy when multiple decisions in gene expression are considered. […] Other mechanistic models have been proposed beyond the demand for increased expression or the accumulation of beneficial mutations (Innan and Kondrashov, 2010), yet do not convincingly resolve the main population genetic dynamical issue”. We have removed the mention to the trade-off here to avoid confusion, now introduced at the beginning of the Results section.

Subsection “Gene duplication helps to better resolve the fitness trade-off”, first two paragraphs;

Rewritten and removed parts related to information transfer. We have also added more details about Figure 3.

Subsection “Expression demand in extreme environments can also lead to the fixation of a newborn duplicate in the population”, second paragraph.

Removed.